# Direct measurements of ice-shelf flexure caused by surface meltwater ponding and drainage

Alison F. Banwell [1,2], Ian C. Willis [1,2], Grant J. Macdonald[3], Becky Goodsell[3] & Douglas R. MacAyeal[3]

Global sea-level rise is caused, in part, by more rapid ice discharge from Antarctica, following the removal of the restraining forces of floating ice-shelves after their break-up. A trigger of ice-shelf break-up is thought to be stress variations associated with surface meltwater ponding and drainage, causing flexure and fracture. But until now, there have been no direct measurements of these processes. Here, we present field data from the McMurdo Ice Shelf, Antarctica, showing that the filling, to ~2 m depth, and subsequent draining, by overflow and channel incision, of four surface lakes causes pronounced and immediate ice-shelf flexure over multiple-week timescales. The magnitude of the vertical ice-shelf deflection reaches maxima of ~1 m at the lake centres, declining to zero at distances of <500 m. Our results should be used to guide development of continent-wide ice-sheet models, which currently do not simulate ice-shelf break-up due to meltwater loading and unloading.

[1] Scott Polar Research Institute (SPRI), University of Cambridge, Cambridge CB2 1ER, UK. [2] Cooperative Institute for Research in Environmental Sciences (CIRES), University of Colorado Boulder, Boulder 80309 CO, USA. [3] Department of Geophysical Sciences, University of Chicago, Chicago 60637 IL, USA. Correspondence and requests for materials should be addressed to A.F.B. (email: alison.banwell@colorado.edu)

Surface meltwater lakes, together with stream and river networks have been widespread on the floating ice shelves that surround the Antarctic continent for decades[1–6]. Lakes are thought to be hazardous to ice-shelf stability by acting as concentrated loads that flex and weaken the floating ice[2–5,7–12]. Surface streams and rivers facilitate the movement of meltwater across ice-shelves[3,6], and therefore have an important role in controlling lake filling and draining[3,5]. When meltwater is simply produced and then re-frozen in-situ, there is no mass change at the surface. Numerical model and laboratory simulations suggest that if meltwater is advected across the ice-shelf surface, causing lakes to fill and drain (resulting, respectively, in local loading and unloading of the surface), the ice-shelf will flex, which may lead to the formation of fractures both within and outside the lake basins[10,13]. A modelling study has suggested that if these fractures intersect other nearby lake basins, a chain reaction of further lake-drainage events may be initiated, potentially contributing to large-scale ice shelf break-up[10]. This scenario is supported by remote-sensing observations of the break-up of the Larsen B Ice Shelf in 2002[1,14]. Ice shelves that have undergone significant thinning due to surface ablation and ocean-driven ablation at the ice-shelf base[15–18] may be particularly vulnerable to break-up[10,14,19].

Although the effects of surface lakes on ice-shelf flexure, fracture and stability have been simulated by small-scale laboratory experiments and theoretical models[9–12,20], there are currently no field data confirming whether ice-shelf flexure (and potential fracture) in response to meltwater movement, ponding or draining actually occurs. Therefore, the aim of the present study was to collect field measurements to examine this process. The key practical constraint on our field campaign was to find a suitable location to collect the data. Given the remoteness of most presently melting Antarctic ice shelves, we undertook our study at the best site available; on the McMurdo Ice Shelf (McMIS) near the logistical hub of McMurdo Station (Fig. 1).

The McMIS is a small (~1500 km$^2$) portion of the Ross Ice Shelf (Fig. 1a). Parts of its surface are covered by large quantities of fine debris[21], most of which was transported into the area by a large ice sheet/shelf system at the Last Glacial Maximum, and is now exposed on the surface due to surface ablation that is balanced by basal freezing of marine ice[22]. This debris gives the surface a low albedo, which facilitates surface and shallow sub-surface melting[23] despite McMIS being relatively far south (~77°). Surface ablation is further enhanced due to the relatively warm prevailing south-westerly winds, which warm adiabatically as they descend onto the McMIS from the nearby landmasses of Minna Bluff and Mount Discovery[22]. Our ~40 km$^2$ study site is located <5 km from the calving front where the ice shelf is relatively thin (10–30 m, refs. [24,25]) (Fig. 1b). Within this area, we measured a mean surface melt rate of 5.2 mm w.e. day$^{-1}$ against 12 ablation stakes (range = 1.0–20.6 mm w.e. day$^{-1}$) from early November 2016 to late January 2017. The high surface ablation rates and shallow ice depths gave us a good chance of measuring significant changes in vertical ice deflection rates in response to variations in surface meltwater ponding and draining.

Here, we present field observations from in and around four meltwater lakes from early November 2016 to late January 2017 (Methods). These observations include data from 12 differential global positioning system (GPS) stations (three per lake site, mounted on poles drilled into the ice, in a ~1–1.5 km transect extending outwards from each lake centre) and three water pressure sensors (one for three out of the four lake sites) (Fig. 1b). Combined, these are the first direct measurements of ice-shelf flexure in response to the filling and draining of surface meltwater lakes. We show that changes in surface lake volumes cause immediate and pronounced ice-shelf flexure; the magnitude of

vertical ice motion decreases as a function of distance from the maximum change in meltwater loading. Our field data are supported by an exact analytic solution for flexure of a floating, thin elastic plate, in which constrained parameter values fall within sensible ranges.

## Results

**Active meltwater lakes versus relict frozen lake scars.** Fieldwork survey corroborated by analysis of satellite-imagery from the previous 18 years suggests that active meltwater lakes on the McMIS form in topographically low areas with high debris concentrations, e.g. Peanut, Ring, Rift Tip and Wrong Trousers (hereafter WT) lakes (Figs. 1b and 2, where at each site, GPS 1 is closest to the lake centre, and GPS 3 is furthest away). The low albedo of the debris, much of which appears to enter the lake basins entrained by inflowing meltwater, enhances melt rates, as does the relatively low albedo of the ponded meltwater compared to the surrounding bare ice[22].

In addition to the active lakes that fill and drain during the melt season, relict, frozen lake scars are also present, and remain almost entirely frozen at their surface year-round due to their relatively high surface albedo. Good examples of these features are the areas where Ring and Peanut GPS 3s are located (Figs. 1b and 2). Low surface melt rates in these areas, combined with vertical hydrostatic adjustments made by the floating ice-shelf, mean that the frozen lake scars appear as raised pedestals compared to the surrounding topography. Meltwater, therefore, often pools to form active meltwater lakes in the topographically-low areas around the pedestalled, relict lake scars (e.g. where Ring and Peanut GPS 1s are located; Figs. 1b and 2).

**Ice-shelf vertical elevation change observations.** At all lake sites during the melt season, the greatest changes in ice-shelf vertical elevation in response to meltwater loading/unloading are close to the active lake centres (i.e. at each site's GPS 1). The data from all four GPS transects support this (Figs. 3 and 4). Data from GPS 1 at all lake sites apart from the Peanut site (where we have no measured water depth data near to GPS 1), also show that there is a clear coincidence between the time periods when lake depths are increasing and when the ice shelf is deflecting vertically downward (Figs. 3 and 4). These temporal coincidences occur immediately before GPS 1 at each lake site reaches its lowest elevation (indicated by a red dot below the red lines, Figs. 3 and 4, top plots), and is clearest at Ring GPS 1 between 21 and 22 December (Fig. 3a, top plot). There is also a clear temporal coincidence between the initiation of vertical uplift at Ring and WT GPS 1s, and the initiation of the Ring and WT lake drainages (Figs. 3a and 4a). The same cannot be said at Rift Tip or Peanut due to missing GPS data at Rift Tip and a lack of water depth data at Peanut (Figs. 3b and 4b).

Data from the Ring site show the most pronounced example of the immediate vertical ice movement in response to the filling and draining of a lake (Fig. 3a). Until mid-December, Ring GPS 1 uplifts slowly at ~3 mm day$^{-1}$, equal to the mean seasonal uplift rate of Ring GPS 2 and 3. This steady uplift is the background trend observed at all 12 GPS stations during the melt season and is likely due to relatively high surface meltwater production and export rates, compared to low meltwater import and ponding rates, causing the ice shelf to uplift hydrostatically as it experiences a net loss of surface mass (see Net meltwater budget calculations section, below, for further analysis and explanation). This upward deflection trend, indicative of net unloading from the area, suggests that the majority of the meltwater that slowly fills Ring lake until mid-December is produced in-situ, as opposed to being transported in from the surrounding area via

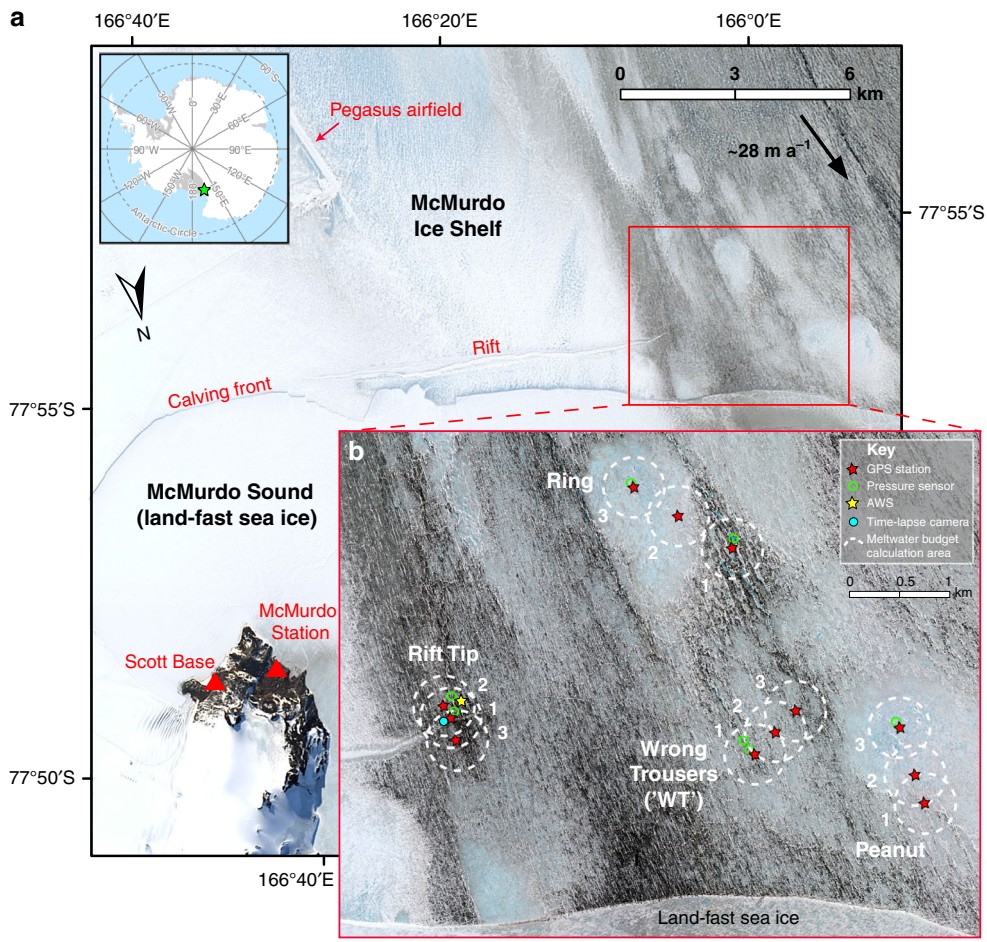

**Fig. 1** Study site on the McMurdo Ice Sheet in the vicinity of McMurdo Station and Scott Base. The background of both **a** and **b** is a WorldView-2 image (©2016, DigitalGlobe) dated 2 December 2016. In **a**, the green star in the top-left inset indicates the location of the McMIS, and the black arrow indicates the local ice flow direction and speed (~335° True at ~28 m a$^{-1}$), based on our own GPS velocity data from the 2016/2017 austral summer. The red box indicates the location of our study site, shown in **b**, in which the four lake sites are labelled. At each lake site, the locations of three GPS stations are marked with red stars (labelled 1 to 3, where 1 is closest to the lake centre and 3 is furthest away). Green open circles mark the locations of pressure sensors; data from three of these are shown in Figs. 3 and 4. A yellow star marks the location of the automatic weather station (AWS), and a blue circle marks the location of a time-lapse camera (used to produce Supplementary Movie 1 and Supplementary Fig. 1). The dashed-white circles of $r = 250$ m, centred on each of the 12 GPSs, show the areas where seasonal meltwater budgets were calculated. (N.B. Rift Tip lake was originally named in a previous (2015/2016) fieldwork season, when it was at the end of a rift. Although this rift has since propagated westwards by ~3 km (ref. [21]), we have kept the original name)

surface melt streams. Thus, although Ring lake fills between late November and mid-December, the dominant trend of Ring GPS 1 is uplift as more mass is being lost from the area near to Ring GPS 1 by meltwater production and export than is being transported into the area and is ponding.

However, from ~14 December, when Ring lake fills more quickly, the dominant trend of Ring GPS 1 is downward deflection, at ~10 mm day$^{-1}$. Between 21 and 22 December, Ring GPS 1 goes down rapidly by ~0.2 m, corresponding to the time when the lake fills most rapidly and reaches >2 m in depth. The downward deflection measured between 14 and 22 December is indicative of net loading in the vicinity of Ring GPS 1, suggesting that inflow of meltwater from the surrounding area is now contributing to the filling of Ring lake. On 22 December, Ring lake starts to drain and Ring GPS 1 immediately starts to rise rapidly by ~50 mm day$^{-1}$. In total, Ring GPS 1 rises by 0.96 m between 22 December and 28 January (Fig. 3a, top plot).

The data from Rift Tip GPS 1 also show a pronounced uplift response to lake drainage (Fig. 3b), but due to missing data (dashed red line), the precise uplift initiation date is unknown. If the uplift response was as fast as it was at Ring, then uplift

initiation was likely to be on or around 15 December, as our Rift Tip water-pressure sensor data (Fig. 3b) and evidence from a time-lapse movie of Rift Tip lake filling and draining (produced from photos taken at 30-min intervals, Supplementary Movie 1) shows Rift Tip lake started to drain on 15 December (Supplementary Fig. 1b). The movie also shows that Rift Tip lake drains via surface overflow in ~9 days, assisted by the incision of a surface stream (Supplementary Fig. 1c), which drains water from ~22 December onwards. As we have no field evidence of hydrologically-induced cracks or moulins, and as the drainage times for the three other lakes was also of the order of days (i.e. longer than the typical time taken for drainage by hydrofracture[26,27]), we assume that they also drain slowly by overflow (e.g. by removal of a natural impediment and/or channel incision at the lake outflow).

**Net meltwater budget calculations.** The net meltwater budgets within circles of radius ($r$) 250 m centred on each of the 12 GPSs (Fig. 1b), each defined as the meltwater ponding volume (calculated from Landsat 8 satellite imagery analysis) minus the

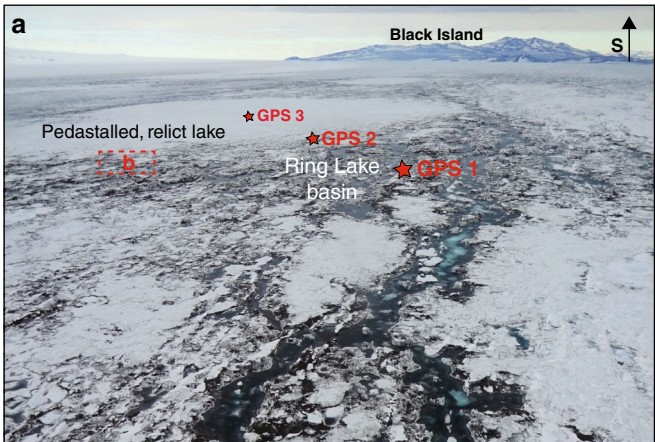

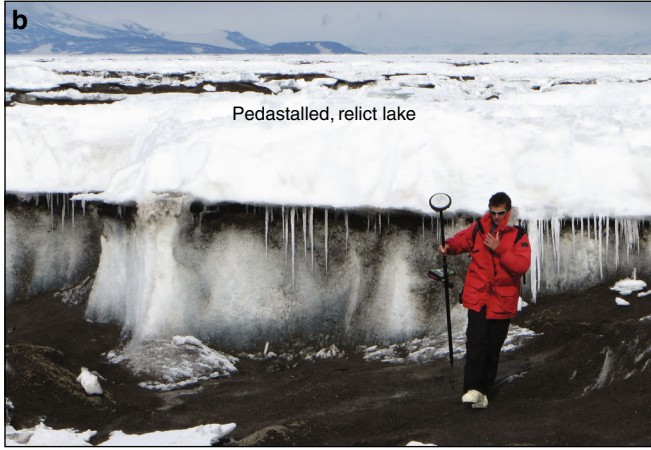

**Fig. 2** Aerial and ground views of the Ring lake site. The aerial view photo in **a** was taken on 18 January 2017, after the lake in the dirty, low topographic area near to GPS 1 had almost completely drained (Fig. 3a, top plot). A pedestalled, frozen lake scar can be seen in the distance (centred on GPS 3). The red box in **a** marks the spot where the photo in **b** was taken. The ground view photo in **b** was taken from the dirty, topographically-low area, looking towards the lake scar in the background, with a person for scale

total meltwater production volume (calculated by a positive degree-day (PDD) model) between early November and late January, were calculated in order to help explain the vertical movements of each GPS station (Figs. 3 and 4 and Methods). Our results show that the seasonal net meltwater budgets at all GPS stations, each defined as the maximum minus minimum net meltwater budget during the 2016/2017 melt season, are negative (Figs. 3 and 4). This is indicative of a net mass loss from each of these 12 areas during the melt season, due to a greater volume of meltwater production (and export), than of meltwater ponding (and import), and explains the dominant uplift signal at all GPS stations through the majority of the melt season (varying between 3 and 10 mm day$^{-1}$ for all GPSs 2 and 3; Figs. 3 and 4). The mass that is lost, i.e. meltwater, is transported either to other areas of the ice shelf or, more likely, to the ocean via surface streams. Some of the measured vertical uplift may be partially attributable to sub-ice-shelf accumulation, but as surface ablation is balanced by sub-ice-shelf accumulation on an annual basis[22], and as surface melting is most prevalent in the summer, we deduce that surface melting is likely the dominant control on ice-shelf uplift change during the summer melt season.

We also note that the short-lived reduction in the uplift rate around 5/6 January in all GPS datasets (Figs. 3 and 4) is likely due to a snowfall event, evidence of which is shown in Supplementary Movie 1. This snowfall event would have temporarily added a small load to the ice-shelf surface and, perhaps more importantly, would have increased the surface albedo (and thereby reduced meltwater production rates) and increased surface meltwater storage in the snow (and thereby reduced meltwater export rates).

For both the Ring and Peanut sites, the locations with the greatest change in the net meltwater budget over the season correspond with the places that experience the greatest total change in vertical ice elevations i.e. at Ring and Peanut GPS 1s (Figs. 3a and 4b). For example, Ring GPS 1, which undergoes the greatest total change in vertical elevation (~1 m) out of all GPS 1s, also experiences the greatest seasonal change in the net meltwater budget in its surrounding area (Fig. 3a). There, the net meltwater budget is greatest ($4.1 \times 10^4$ m$^3$) on 24 December, coinciding with the time when Ring GPS 1 is at its lowest elevation, and smallest ($-1.6 \times 10^4$ m$^3$) on 18 January, when Ring GPS 1 is close to its highest elevation. The net removal of water from the $r = 250$ m circular area centred on Ring GPS 1 between those dates is $5.7 \times 10^4$ m$^3$ (Fig. 3a); a volume that we refer to as the seasonal net meltwater budget. It is not possible to identify a similar

correspondence between the maximum changes in net meltwater budget and ice elevation at the Rift Tip and WT sites (Figs. 3b and 4a). However, for both the Rift Tip and WT GPS transects, the seasonal changes in net meltwater budgets are more similar across each transect (i.e. between the three GPS stations), than they are at the Ring and Peanut sites.

**Analytic expression for idealised ice-shelf flexure.** Compared to the GPS 1 station data, which record pronounced vertical movement at the centre of all lake sites, GPSs 2 and 3 are distal from the lake centres and generally do not show a pronounced response to lake filling or drainage (Figs. 3 and 4). The exception to this is Rift Tip GPSs 2 and 3, which are both <250 m of Rift Tip GPS 1 and uplift vertically at a rate that is almost as rapid as that measured at GPS 1 when Rift Tip lake drains (Fig. 3b). Therefore, the data from all sites suggests that the flexural responses to load changes are local (e.g. <500 m from lake centres). An exact analytic solution for the removal of a disk-shaped load from a floating, thin elastic-plate[10,11] (Methods) supports this observation. For example, using a combination of constrained parameter values that produce the best match between the analytic solution and measured deflection at Ring lake's centre (for more detail, see below, and Methods), its centre is simulated to rise by ~1 m, whereas the ice surface >250 m away from the lake centre rises by <10 mm (Fig. 5a). Our field measurements at Ring GPSs 2 and 3 record some net uplift during the season (80 and 70 mm, respectively), but this is due to the slightly negative seasonal net meltwater budget at each of these locations (Fig. 3a).

The parameters in the analytic solution deemed to be most sensitive are lake radius ($R$); effective ice thickness ($H$); and Young's Modulus ($E$). Sensitivity tests were run for varying combinations of values for these parameters within the following ranges: $R = 50–250$ m; $H = 10–30$ m; and $E = 1–10$ GPa. These ranges are guided by our field data ($R$); McMIS data collected by others ($H$)[24,25]; and, values derived through modelling and laboratory experiments by others ($E$)[10,11,28]. For the Ring lake site, calculated centre-lake deflection and maximum von-Mises stress for varying combinations of values for the parameters $R$, $H$ and $E$ are given in Supplementary Table 1. Optimal values of these parameters for the Ring lake site, where our net meltwater budget calculations show that a meltwater volume of $5.7 \times 10^4$ m$^3$ is removed during the melt season, are found to be 125 m, 10 m, and 1 GPa, respectively (Fig. 5a), giving the ~1 m uplift at Ring

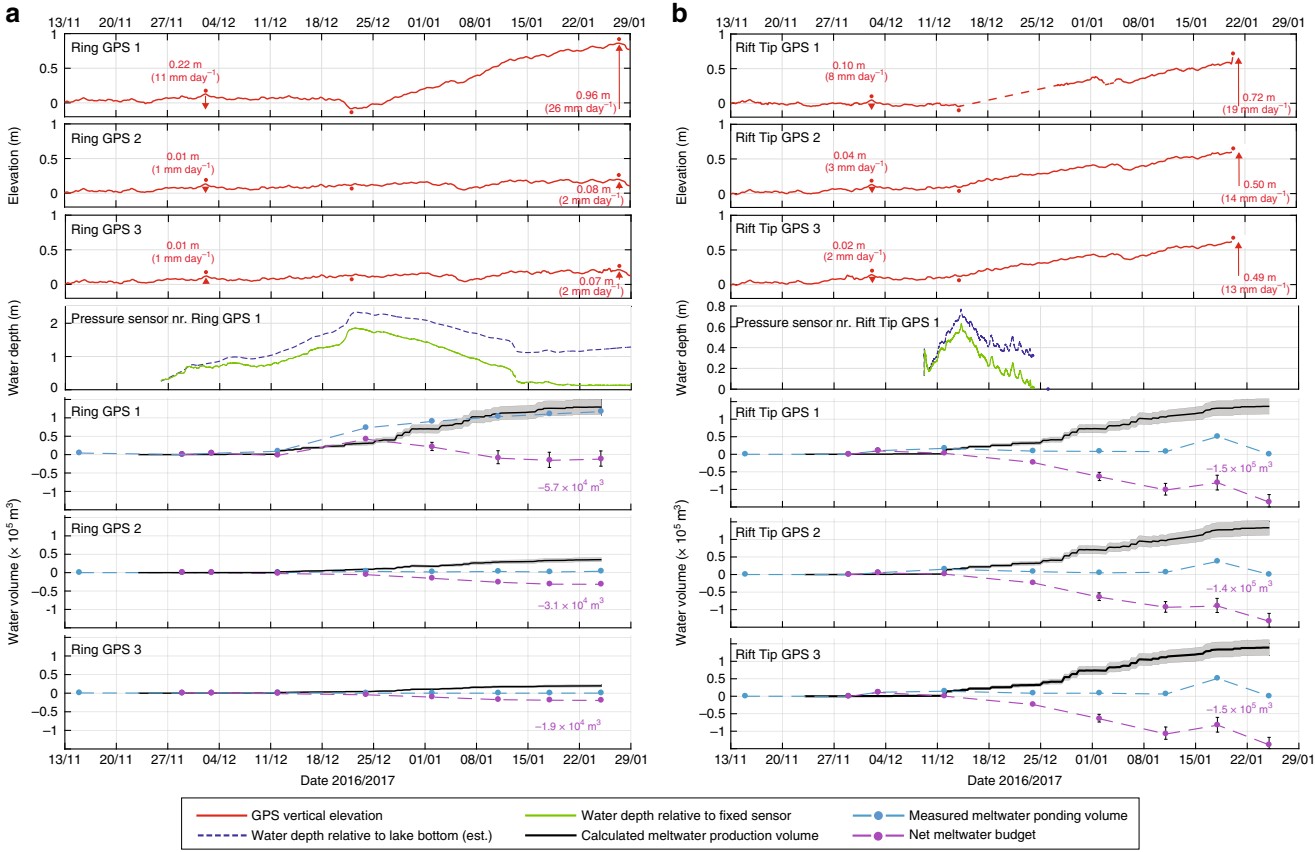

**Fig. 3** Vertical ice shelf displacement with water depths and seasonal meltwater budgets for Ring and Rift Tip lake sites. **a** is for Ring, and **b** is for Rift Tip. For each site, the top three plots show the vertical ice-shelf displacement (relative to arbitrary elevation) from three GPS stations (Fig. 1b, red stars). The first red dot above each time series shows when GPS 1 reaches its highest elevation, before reaching its lowest elevation, shown by a second red dot. The third red dot indicates when GPS 1 reaches its highest elevation during the whole time period. Numbers next to red arrows (which depict the direction of movement) are the total vertical deflections, and deflection rates, measured by each GPS over the two respective time periods between the first and second red dots, and the second and third red dots. Some of the elevation data for Rift Tip GPS 1 are missing so are linearly interpolated (dashed-red line). The fourth plot for each lake site shows water depth data from the pressure sensor nearest to each GPS 1 (Fig. 1b, green open circles). Water depths relative to lake bottom were estimated from sensors fixed at constant heights (see Methods). The bottom three plots at each site show the calculated seasonal meltwater budgets around each GPS station. The black lines (and grey shading) show cumulative volumes (and errors) of meltwater production calculated by a PDD model; light blue dots show the measured volumes of meltwater ponding from 9 cloud-free Landsat 8 images (see Supplementary Table 2 for image dates), and the dashed-light blue lines show these data linearly interpolated between the image dates; purple dots (and whiskers) show the net meltwater budget (and errors) on each image date, and the dashed-purple lines show these data linearly interpolated between the dates. Purple numbers next to the net meltwater budget plots refer to the seasonal net meltwater budgets, each defined as the maximum minus minimum net meltwater budget during the melt season, which are all negative. See Methods for details of GPS and water-depth data processing, and meltwater budget and error calculations

lake's centre that is consistent with our GPS 1 measurements (Fig. 3a).

The optimal values of *H* (10 m) and *E* (1 GPa) established for the Ring lake site were then used within the analytic solution and applied to the other three lake sites to see if the analytic solution agreed with GPS 1 vertical elevation data measured there. The other lakes have different meltwater unloading inputs, as indicated by our seasonal net meltwater budget calculations (Figs. 3 and 4, lower three plots in each), thus we expected that changing the value of *R* for each lake would be necessary. Agreement between the simulated and measured centre-lake deflections was obtained with *R* values of 250, 175 and 200 m for Rift Tip, WT and Peanut lake sites, respectively (Supplementary Fig. 2), which are plausible based on our field and satellite-based observations. With these parameter values, the ice surface is simulated to rise by <10 mm at distances of >370, >284 and >302 m away from the centres of Rift Tip, WT and Peanut lakes, respectively. Therefore, as with the results for the Ring lake site, the results for the other three lake sites show that meltwater loading/unloading has only a local flexural effect (Supplementary Fig. 2). At Rift Tip, GPSs 2 and 3 are <370 m away from Rift Tip lake centre, so some of their observed uplift is explained by the net meltwater unloading at GPS 1, with the rest due to meltwater unloading closer to those stations (Fig. 3b, Supplementary Fig. 2a). Similarly, WT and Peanut GPS 2s are, respectively, <284 and <302 m away from their respective lake centres, but their GPS 3s lie further away. The measured deflections at their GPS 3s are due entirely to the net meltwater unloading close to those stations; at their GPS 2s it is due mainly to local unloading, with a small fraction in response to unloading at their GPS 1 sites (Fig. 4a, b, Supplementary Fig. 2b and c).

## Discussion
While lake filling and, more significantly, drainage, caused differential changes in local ice-shelf elevation, and therefore ice

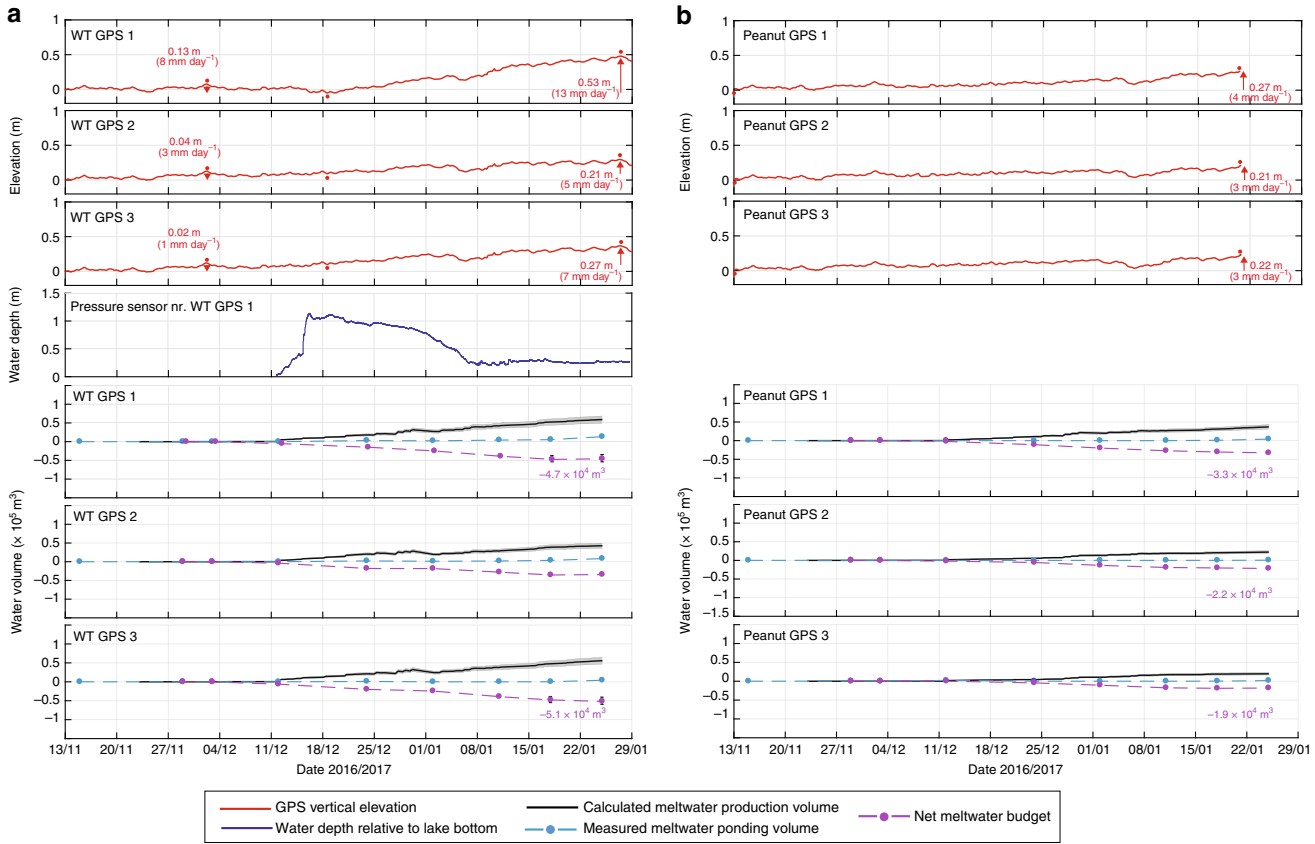

**Fig. 4** Vertical ice shelf displacement with water depths and seasonal meltwater budgets for WT and Peanut lake sites. **a** is for WT and **b** is for Peanut. For each site, the top three plots show the vertical ice-shelf displacement (relative to arbitrary elevation) from three GPS stations (Fig. 1b, red stars). The first red dot above each time series shows when GPS 1 reaches its highest elevation, before reaching its lowest elevation, shown by a second red dot. The third red dot indicates when GPS 1 reaches its highest elevation during the whole time period. Numbers next to red arrows (which depict the direction of movement) are the total vertical deflections, and deflection rates, measured by each GPS over the two respective time periods between the first and second red dots, and the second and third red dots. The fourth plot for the WT site shows water depth data from a pressure sensor nearest to GPS 1 (Fig. 1b, green open circles). The bottom three plots for each site show the calculated seasonal net meltwater budgets around each GPS station. The black lines (and grey shaded areas) show cumulative volumes of meltwater production (and errors) calculated by a PDD model; light blue dots show the measured volumes of meltwater ponding from 9 cloud-free Landsat 8 images (see Supplementary Table 2 for image dates), and the dashed-light blue lines show these data linearly interpolated between the image dates; purple dots (and whiskers) show the net meltwater budget (and errors) on each image date, and the dashed-purple lines show these data linearly interpolated between the dates. Purple numbers next to the net meltwater budget plots refer to the seasonal net meltwater budgets, each defined as the maximum minus minimum net meltwater budget during the melt season, which are all negative. See Methods for details of GPS and water-depth data processing, and meltwater budget and error calculations

flexure, across all four of our lake sites, no observable fractures formed. It is reasonable to expect that hydrofracture does not assist flexure-driven fracture when ice-shelf thicknesses are small, because the hydrostatic head can never become large enough[29]; and/or lakes on the McMIS simply do not currently reach large enough volumes to produce sufficient tensile-stress levels for fracture initiation[10,30]. Small lake volumes may be attributable to the low amplitude of surface topographic undulations, which means that lakes drain via surface overflow after reaching just a small water volume. An extensive stream/river system may help lakes to drain by overflow, and may also intercept and evacuate meltwater before it is ever able to enter potential lake basins. Much of this meltwater will be transported off the ice shelf and into the ocean[31,32,33], thereby contributing to the ice shelf's negative seasonal net meltwater budget. Alternatively, local flexure-induced tensile stresses alone (Fig. 5b, Supplementary Fig. 2) may be sufficiently large for fracture, but stresses from further afield, such as back-stresses from land-fast sea ice[34] and/ or stresses from larger-scale ice-flow[35], may be acting to prevent fracture initiation[5].

Although our results indicate that there is currently a negative net meltwater budget on the McMIS during the melt season, melt rates on many of Antarctica's ice shelves are expected to increase two- to three-fold by 2050[36]. Therefore, meltwater volumes may become so great that even if large-scale river systems develop to evacuate meltwater off the ice shelves and into the ocean[31,32,33], their discharge capacity may be insufficient to prevent positive net meltwater budgets developing during a melt season (or over successive seasons). Such increased meltwater loading may increase the potential for fracture formation, and ultimately, ice-shelf break-up, perhaps once a given ponding density threshold is exceeded[10]. Surpassing a given ponding density may also enable a chain-reaction style lake drainage process to assist with ice-shelf break-up[10]. Currently on the McMIS, surface lakes are not sufficiently widespread for the localized meltwater-loading induced flexure (and potential fractures) to affect more than one lake. Other ice shelves that are already experiencing more widespread melting and pond formation may be more vulnerable to break-up[5].

The results of this study, which are based on field measurements, show that surface meltwater ponding and drainage has a

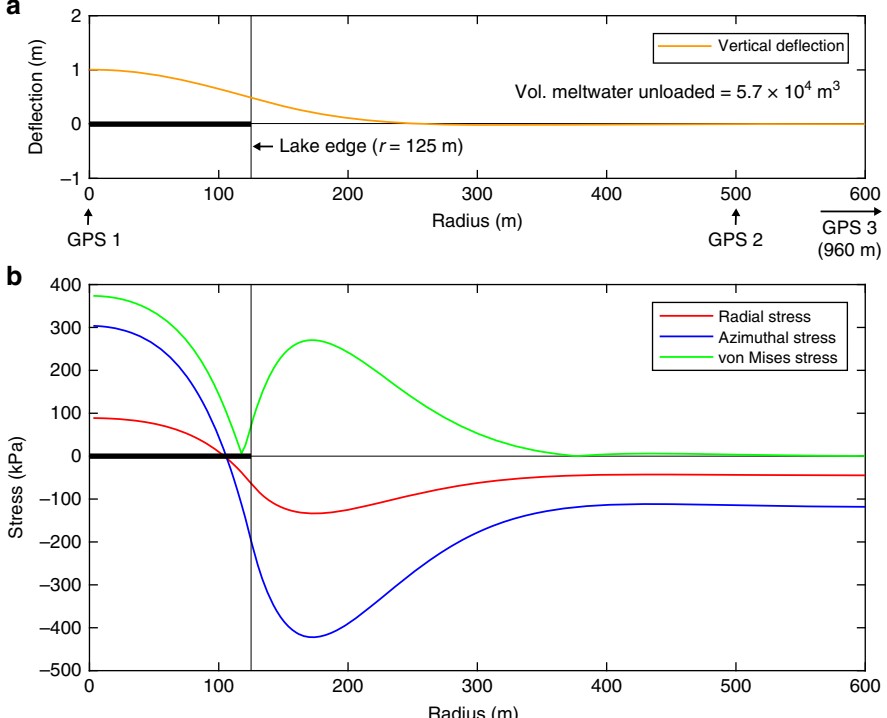

**Fig. 5** Vertical ice shelf deflection and stresses associated with the drainage of Ring Lake. **a** Vertical ice-shelf deflection (orange line), as a function of distance from lake centre, is computed by an exact analytic solution for thin elastic plate flexure above seawater, in response to the drainage of Ring lake (of volume $5.7 \times 10^4$ m³, equal to the calculated seasonal net meltwater budget within a 250 m radius of Ring lake GPS 1, Fig. 3a). This result shows the best match between the analytic solution and the measured deflection at Ring lake's centre (i.e. GPS 1) and uses the following parameter values: $R = 125$ m, $H = 10$ m, and $E = 1$ GPa. The locations of GPSs 1, 2, and 3 (960 m from the lake centre) are indicated. The associated stresses (radial (red line), azimuthal (blue line), and von-Mises (green line) as a function of distance from lake centre), shown in **b**, are all evaluated at the upper ice-shelf surface

prominent and instantaneous effect on ice-shelf vertical deflection and flexure. The magnitude of the vertical ice-shelf deflection decreases from maxima of ~1 m at the centres of the maximum load changes to zero at distances of <500 m; observations that are supported by an exact analytic solution for flexure of a floating, thin elastic plate, in which constrained parameter values fall within sensible ranges.

Ultimately, the observations presented here provide an initial performance constraint to guide the development of regional- and continent-scale ice-sheet models[37,38] to produce more accurate predictions of future sea-level rise following ice-shelf collapse[7,39–41]. The process of ice-shelf flexure in response to surface meltwater load changes[11], which may lead to fracture[9], is not yet simulated by these models.

## Methods

**GPS deployment and data processing.** Twelve differential GPS stations were deployed in the field area from mid-November 2016 to mid-January 2017 (Figs. 1b and 2, red stars). The GPS stations, provided by UNAVCO, were Trimble NetR9 with Zephyr Geodetic antennas, with photovoltaic power supplies. The antennas were placed on 3 m aluminium poles driven ~2.5 m (with a Kovacs drill) into the ice at the start of the field campaign, leaving the antennas initially elevated ~0.5 m above the ice surface. At the end of the campaign, surface ablation caused some pole emergence at the sites, however all antennas were still rigidly fixed with respect to the subsurface ice reference frame at the bottom of the survey poles, and the poles had tilted <5°.

The L1 and L2 carrier frequency data (15 s sample rate) recorded by the GPS receivers were processed using the TRACK software package maintained by the Massachusetts Institute of Technology using the GPS base station located at McMurdo Station (~15 km from the field area, and operated by UNAVCO). Parameters specifying standard deviation for horizontal and vertical motion used by the Kalman filter within TRACK, 30 mm of motion between samples in each direction, were estimated conservatively, and variation of this choice was not found to change the processed data significantly. Following the application of TRACK to

determine the vertical elevation time series, the data were quality checked for outliers and obvious cycle slips (10 s of mm displacement over time periods of <10 s of minutes), and these were replaced with interpolations.

Tidal displacement showing the dominant diurnal tide of McMurdo Sound with a weak semidiurnal component and the familiar spring-to-neap tidal cycle was the principal vertical motion in the time series. To restrict attention to only long-period monotonic vertical displacements associated with load-driven ice-shelf flexure, the tidal signals from each of the 12 stations were removed. This was done using a least-squares estimation of pure sinusoidal variation within each time series, applied with 3 dominant diurnal and 4 dominant semidiurnal frequencies (K1, O1, S1, M2, S2, N2, K2) to produce the long-term vertical displacement residual presented in this paper. Subsequently, the data were smoothed using a 24-h running mean to remove other lower and higher frequency signals, including non-sinusoidal signals that are nearly at the tidal frequencies (e.g. diurnal multi-path error).

Following de-tiding and smoothing, the long-term vertical displacement residual was corrected for the inverse barometer effect (IBE) using barometric data (logged every 10 min) from two AWSs located 10–15 km from the field area (operated by the University of Wisconsin Antarctic Meteorological Research Center (AMRC), Pegasus North and Willy Field). The vertical displacement coefficient for the IBE used was 0.9948 cm mb⁻¹. Because the IBE is common to all stations, errors in its application have no effect on vertical displacement differences between stations used to infer ice-shelf flexure.

Applying the time-averaging and tide removal algorithms described above to the elevation time series presented in Figs. 3 and 4 renders errors due to short-term (seconds to minutes) influences (due to GPS receiver effects and data processing uncertainties, and to ice-shelf movements due to long-period ocean swell) to ±10 mm; and renders errors due to long-term (hours to days) effects (due to ocean tide and the IBE) to ±50 mm. To arrive at these estimates of uncertainty, we computed the standard deviation of vertical-elevation differences between pairs of stations in hourly time-windows over 1944 windows in an 81-day period when all stations were recording data simultaneously. The histogram of standard deviations for the 1944 h-long time windows peaked at 48 mm, suggesting that the standard deviation of errors affecting our analysis is <50 mm. Figures 3 and 4 show 24-h running means of relative vertical GPS displacement. We expect the error associated with these running means to be $1/(\sqrt{24})$, ≈1/5 of the 50 mm estimate derived from the hourly windows, or ~±10 mm. It does not make sense to plot this small error bracket on all of our GPS time series as they would not be visible.

**Pressure sensor deployment and data processing**. Eight HOBO® (U20L-01) water pressure sensors, with a 0–9 m range and a 10-min logging interval, were deployed in and around the lake study sites from mid-November 2016 until mid-January 2017 (Fig. 1b, open green circles). We show water-depth data from the three sensors that were retrievable and which showed non-negligible water depth changes (Figs. 3 and 4). These were the sensors located nearest to Ring GPS 1, WT GPS 1 and Rift Tip GPS 1.

To account for barometric pressure fluctuations, the data were corrected against air pressure data from the AMRC Pegasus North AWS. The sensors have a typical water level accuracy of ±3 mm and a resolution of ~2 mm (http://www.onsetcomp.com/products/data-loggers/u20l-01). 

At all sites, we initially deployed two sensors at the height of the ice surface, a fixed sensor secured to an aluminium pole drilled into the ice, and a loose sensor which was to slide down the pole as the ice ablated. Following ref. [42], we had hoped to calculate lake-bottom ablation using this method. Here though, we just report water depths relative to the ablating lake bottom. At WT GPS 1, the sensor successfully slid down the pole as the surface ablated. It therefore measured the water depth relative to the lake bottom, through time (Fig. 4a, dark blue line). At Ring and Rift Tip GPS 1s, however, we only have data from the height of the fixed sensors (Fig. 3a, b, green lines) because the loose sensors were either irretrievable (at Ring GPS 1) or remained stuck against the pole at the height of the fixed sensor (at Rift Tip GPS 1). At both Ring and Rift Tip GPS 1s, we therefore estimate lake depth relative to the lake bottom using the following procedures. The fixed sensors were initially secured to their pole at the lake bottom, where there was 0 m water depth. At Ring GPS 1, we measured the water depth between the fixed sensor and the lake bottom (1.3 m) when we retrieved the sensors. Using these data, we calculated the average lake-bottom ablation rate at Ring GPS 1 during the sensor's deployment to be 20.3 mm day$^{-1}$, which we used in conjunction with the fixed sensor water depth data (Fig. 3a, green line) to estimate water depth relative to the lake bottom (Fig. 3a, dashed dark blue line). At Rift Tip GPS 1, the lake bottom was dry when we retrieved the sensors on 18 January. As we were unable to calculate an ablation rate for the time the lake contained water, we used the ablation rate measured at Ring GPS 1 of 20.3 mm day$^{-1}$ to correct the fixed sensor data to produce a time series of estimated depth relative to the lake bottom (Fig. 3b, dashed dark blue line).

**Time-lapse camera deployment and analysis**. A Harbortronics Cyclapse time-lapse camera system, containing a Canon EOS Rebel T6i, was deployed from 25 November 2016 to 27 January 2017 beside Rift Tip GPS 1 (Fig. 1b, filled blue circle). It was mounted on a steel pole attached to a 4-in. × 4-in. wooden stake drilled into the ice and it was programmed to take a photo every 30 min. Following retrieval, the 3015 images were date- and time-stamped in MATLAB (three of which are shown in Supplementary Fig. 1), then processed in GoPro Studio to produce a movie with a frame rate of 30 s$^{-1}$ (Supplementary Movie 1).

**Seasonal net meltwater budget calculations**. The 'net meltwater budget', defined as the total meltwater ponding volume minus the total meltwater production volume, was calculated within circles of radius ($r$) = 250 m (area = ~2 × 10$^5$ m$^2$), centred on each of the 12 GPSs (Fig. 1b, white dashed circles) through the 2016/2017 melt season. Circles with this dimension were chosen for various reasons. First, the maximum distance between any two GPS stations along each lake transect is 500 m, so 250 m radii circles around those stations do not overlap. Second, the frozen lake scars in our two partially clean lake sites, Peanut and Ring, have radii of ~250 m, and thus it did not seem sensible to make the circles for meltwater budget calculations any smaller. And third, using larger circles would have resulted in a significant overlap between the circles, and therefore, potentially similar meltwater budgets around each GPS station. We note that sensitivity tests showed that changing the areas of the circles that were analysed (from $r$ = 125–500 m) did not change the meltwater budget trends for any of the GPS. 'Seasonal net meltwater budgets' at all GPS stations were also calculated, each defined as the maximum minus minimum net meltwater budget during the 2016/2017 melt season. Details of meltwater ponding and meltwater production calculations are given below.

**Observed surface meltwater ponding calculations**. We used data collected by the Landsat 8 sensor to estimate areas and depths, and therefore volumes, of ponded meltwater over our study region through the 2016/2017 melt season. Landsat 8 was launched in 2013 and hosts the operational land imager (OLI) spectrometer, suitable for lake area and depth estimation[33,43]. Only images with no cloud-cover across our four lake study sites were used. In total, 10 such images between 1 November 2016 and 31 January 2017 were available (Supplementary Table 2).

Reflectance values were used to extract both the area and the depth of surface water using a combination of bands. Landsat 8 bands 2 (blue, 450–510 nm), 4 (red, 640–670 nm), 7 (shortwave infrared, 2100–2300 nm) and 8 (panchromatic, 500–680 nm) were cropped to our area of interest (using Extract by Mask in ArcMap$^{TM}$). We used each image's metadata to convert digital numbers to top-of-atmosphere (TOA) reflectance and to correct for solar elevation. These TOA reflectance values represent an adequate proxy for surface reflectance[43].

To identify water-covered pixels, we calculated the normalized difference water index adapted for ice (NDWI$_{ice}$), defined as:

$$NDWI_{ice} = (B2 - B4/B2 + B4) \qquad (1)$$

where B2 and B4 represent the blue and red bands, respectively. Owing to the spectral dependency of water reflectance, pixels covered in deeper water will have higher NDWI$_{ice}$ values. Pixels with NDWI$_{ice}$ > 0.07 were assumed to be water-covered. This value is lower than that (0.12) used to detect water-covered pixels in Landsat 8 in other studies[32,44], because we lowered this threshold by the minimum amount necessary (in increments of 0.01) to include pixels that we knew to be water-covered from our ground instrumentation (i.e. our 8 water pressure sensors). Nine images (15 November 2016–25 January 2017 inclusive) were found to contain water-covered pixels (Supplementary Table 2).

To calculate water depth for the pixels identified as being water-covered, we employed the physically-based, single-band, water-depth retrieval algorithm originally based on the Bouguer–Lambert–Beer law[45,46], which describes the attenuation of radiation through a water column; deeper water results in higher light attenuation within the column than shallower water. The expression for reflectance immediately below the water surface for optically shallow, homogeneous water, $R(0-)$, is given by

$$R(0-) = R\infty + (A_d - R\infty)\exp(-gz) \qquad (2)$$

where $A_d$ is the lake bottom albedo, $R\infty$ is the reflectance of optically deep water, and the coefficient $g$ accounts for losses in upward and downward travel through the water column and varies with the wavelength used[43]. Solving this equation for water depth ($z$) gives:

$$z = [\ln(A_d - R\infty) - \ln(R_{water} - R\infty)]/g \qquad (3)$$

where $R_{water}$ is the reflectance of a water-covered pixel. $A_d$ was calculated image-by-image on a pixel-by-pixel basis, by taking the mean reflectance of a ring dilated by one pixel around each ponded region[47,48], an improvement on previous studies that used static values across a region[46,49]. Some images did not contain optically deep water, and for those images, the difference between using an $R\infty$ value of 0 and a value obtained from the ocean was negligible; for these reasons, an $R\infty$ value of 0 was used. The depth of each water-covered pixel was calculated by averaging the water depths ($z$) derived from Landsat 8's bands 4 and 8 (ref. [43]). The values for $g$ (0.7507 for band 4 and 0.3817 for band 8) were taken from ref. [43]. This method of calculating water depth makes several assumptions, including the lake substrate is homogenous and the impact of any dissolved matter in the water on absorption is negligible; and there is no scattering of light from the lake surface associated with roughness due to wind[46]. We appreciate that the first assumption may be particularly problematic in our study area due to the large amounts of fine debris that accumulates in the areas where water ponds.

Finally, after water depth was calculated, dry debris that was falsely identified as water, was removed from the images. As dry debris is known to have high band 7 and low band 2 reflectance values (whereas water has low band 7 and high band 2 reflectance values), this was done by masking out band 7 pixels with reflectance >0.4 and band 2 pixels with reflectance <0.2. These thresholds were determined by performing a combination of visual analysis and inspection of the reflectance values in bands 2 and 7 for water and debris. A blue-band threshold has similarly been employed in other studies to remove pixels falsely identified as water[50,51], but we are not aware of the shortwave infrared band having previously been used for a similar purpose.

We do not have errors on our meltwater volume calculations as our method for determining a suitable NDWI$_{ice}$ threshold (see above) is assumed to produce negligible errors in the calculation of water-covered pixel areas; and the average depth error across multiple water-covered pixels using bands 4 and 8 has been shown to be zero (because positive and negative errors cancel out)[43].

**Surface meltwater production calculations**. We used a PDD model to calculate meltwater production during the 2016/2017 melt season. Two PDD factors, one for dirty, low-albedo, ice, and one for clean, high-albedo, ice, were derived empirically from our 2-m air temperature data and in-situ ablation measurements. Air temperature data were logged every 15 min at our AWS, which was installed from 22 November 2016 to 27 January 2017 in the centre of Rift Tip lake (Fig. 1b, yellow star). Surface ablation was measured against each of the 12 GPS antenna poles, between the time of their installation in November 2016, and their retrieval in late January 2017. The 12 measurements clearly split into two groups; a dirty ice group (Rift Tip GPSs 1 and 2, and WT GPS 1) with melt rates >12 mm w.e. day$^{-1}$, and a clean ice group with melt rates <4 mm w.e. day$^{-1}$. Our dirty ice and clean ice PDD factors are the mean of the PDD factors for the three dirty stake locations (47.5 mm w.e. °C$^{-1}$ day$^{-1}$) and the 9 clean locations (6.4 mm w.e. °C$^{-1}$ day$^{-1}$), respectively. The Standard Errors of these means are 9.0 and 0.9 mm w.e. °C$^{-1}$ day$^{-1}$, respectively, i.e. 19% and 13% of the respective PDD factors.

To apportion pixels within each circle around each GPS into two categories, dirty and clean, we applied the MATLAB function 'graythresh' to the band 8 TOA reflectance values from the nine Landsat 8 images for the 2016/2017 melt season that our previous analysis had shown to include surface meltwater (Supplementary

Fig. 3 and Supplementary Table 2). Using Otsu's method, this function chooses the threshold value that minimizes the intra-class variance of the black and white pixels. This dynamic thresholding approach is empirical, and justified by the observation that dirty pixels are darker than clean pixels at wavelengths in the solar reflective part of the spectrum. Dynamic thresholding has been used in a variety of previous studies[52] to categorise MODIS images into water and not water-covered areas. We note that it was not possible to simply choose a threshold by eye as the histograms of TOA reflectance were generally not indicative of a bimodal distribution. Once chosen, the threshold was used to produce a binary image of clean (i.e. white) and dirty (i.e. black) pixels and, for quality control, they were visually inspected and compared to the true colour Landsat 8 images and histograms (Supplementary Fig. 3).

From these binary images, the percentage of dirty versus clean pixels in each circle of each image was calculated (Supplementary Fig. 3). As daily volumes of calculated meltwater production through the season were required, we linearly interpolated these percentage values to daily values between each of the nine image dates. Finally, for each day of the 2016/2017 melt season, the clean and dirty PDD factors were applied proportionately to each circle, and daily volumes of total meltwater production were calculated. To calculate the errors for the volumes of meltwater production, we assumed that the errors associated with the apportioning of dirty versus clean pixels were negligible, and used the mean Standard Error of the means of two PDD factors (16%).

**Exact analytic solution for flexure of a floating thin elastic plate**. Flexure of an ice shelf subject to changing surface-meltwater loads can be represented, for the purpose of estimating stress magnitudes, using an analytic solution based on thin elastic (Kirchhoff) plate theory[11,53]. Following ref. [10], we use an azimuthally-symmetric solution valid for $r > 0$ to the thin-elastic-plate flexure equation (also known as the Kirchhoff–Love equation, but modified to account for buoyancy associated with ocean water below the thin plate) in which disk-shaped meltwater loads, or anti-loads (associated with the drainage of lakes), are confined within a region $r \leq R$ using polar coordinates $r$, $\theta$, and where $R$ is the radius of the lake or drained-lake. The vertical displacement of the elastic plate, $\eta(r)$, for $0 \leq r \leq \infty$, is expressed in terms of Kelvin–Bessel functions (as derived by ref. [54]), and displayed in ref. [11]. An undeflected ice shelf at a large distance (i.e. $r \approx \infty$) is assumed, which is appropriate because the grounding lines and ice front are $\geq 2$ km from all lake centres.

Values for the three parameters: lake radius ($R$); effective ice thickness ($H$); and Young's Modulus ($E$), were varied to produce the best match between the analytic solution and measured lake-centre deflection for Ring site (Fig. 5; Supplementary Table 1). Having identified suitable parameter values for $H$ and $E$, these were held constant, while $R$ was varied (by 25 m increments) to find the best match between the analytic solution and measured lake-centre deflection for Rift Tip, WT and Peanut sites (Supplementary Fig. 2). Following ref. [10], the other parameter values are kept constant in this study: $\rho_{sw} = 1028$ kg m$^{-3}$ is the density of seawater, $\rho_{ice} = 910$ kg m$^{-3}$ is the density of ice, $\rho_{fw} = 1028$ kg m$^{-3}$ is the density of fresh water, $g = 9.81$ m s$^{-2}$ is the acceleration of gravity, and $\nu = 0.3$ is the Poisson ratio. For simplicity, and because debris layers were relatively thin (order 10 s of cm), we did not account for changes in the effective density of ice due to debris content. Input to the analytic solution is a disk-shaped meltwater load removal, taken as the calculated seasonal net meltwater budget for each site (Figs. 3 and 4).

We use an elastic treatment of flexure stresses induced after lake drainage, under the assumption that the timescale of lake drainage via overflow is relatively short compared to the Maxwell time[10,11]. However, in reality, as it may take a year or more for a lake to fill, a viscous response of the ice shelf will be present too[12,20,53]. The maximum flexure stresses implied by the elastic solution therefore represent maximum upper bounds; with a viscoelastic model[12,20], modelled deflections at all GPS stations, including the lake centre stations, would be lower, and the flexure stresses would also all be lower. Additionally, using a viscoelastic model, the ice-shelf flexural response to the change in lake load would be even more local than that observed with the current analytic solution for flexure of a thin elastic plate, which our GPS data would support.

## Data availability

The field-derived GPS (https://doi.org/10.15784/601107) and AWS (https://doi.org/10.15784/601106) data are archived at the USAP Data Center. Landsat 8 tiles can be obtained from Earth Explorer (http://earthexplorer.usgs.gov/).

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

## Acknowledgements

This fieldwork component of this paper was supported by the U.S. National Science Foundation under award PLR-1443126 to the University of Chicago. Additionally, A.F.B. was supported by a Leverhulme Early Career Fellowship (ECF-2014-412) and a CIRES Post Doctoral Visiting Fellowship. I.C.W. was supported by a CIRES Sabbatical Visiting Fellowship, and G.J.M. was supported by a NASA Earth and Space Science Fellowship (NNX15AN44H). The authors are grateful to the numerous people associated with the U.S. Antarctic Program, and the staff of McMurdo Station, who helped to make the field campaign possible. The authors thank UNACVO, particularly Brendan Hodge, who helped to deploy the GPS systems, scientists at the Polar Geospatial Center, notably Mike Cloutier, who provided satellite imagery (including the WorldView-2 image presented in Fig. 1) and David Mayer at USGS, who provided invaluable GIS help. Finally, the authors thank University of Colorado Boulder Libraries Open Access Fund for partially funding this publication.

## Author contributions

A.F.B. led the field project and the preparation of the manuscript. I.C.W. ran the PDD melt model. G.J.M. calculated meltwater ponding volumes and determined brightness thresholds for dirty versus clean ice from satellite imagery. D.R.M. processed the GPS data. A.F.B. and D.R.M. implemented and ran the analytic thin elastic plate over seawater solution. All authors were involved in the planning and execution of the field campaign and in the drafting and editing of the manuscript.

## Additional information

**Competing interests:** The authors declare no competing interests.

