## [Peer Review File · Nature Communications]

Reviewers' comments:

Reviewer #1 (Remarks to the Author):

Surface meltwater movement, ponding and drainage causes ice-shelf flexure

The manuscript presents field-based data relating to the vertical surface movement of the Ross Ice Shelf at different distances from lake basins that fill and empty, demonstrating differential surface uplift and lowering. These data are also linked to a model of ice-shelf flexure. The data are hard-earned and deserve publication. However, that the ice shelf flexes in response to differential surface loading is not a particularly novel finding as it follows directly from physical considerations and has consequently been assumed by previous studies. Nonetheless, this manuscript does present the first direct measurements of such flexure (to my knowledge). However, these measurements are themselves not perfect, being driven by fairly small meltwater fluxes and potentially being influenced by the ice shelf margin. One particularly interesting aspect of the manuscript that is currently, in my opinion, underrepresented is the integration of these field data with a computer-based model of flexure response. The principal thrust of the rationale stated for the research is that these data are needed in order to contribute to models of ice shelf behaviour (lines 36 to 38 for example). As I see it, the manuscript potentially has this capacity but does not currently elevate the model structure or calibration and validation to the status that it should have to address this ambition. I would encourage the authors to recast the manuscript such that the field data are used explicitly to derive parameter and variable values for the model, allowing the model to be refined and presented in such a way that the process can be integrated into larger-scale models. In this way, I feel the novel value of the manuscript will be made clearer and the work will be more directly aligned to its rationale.

Some more specific comments follow:

Location/Line	Comment/Suggestion
Title	I think this could be written more specifically to reflect the novel advances of the research presented in this manuscript. That ponding and drainage causes flexure is not debated and has been assumed in the past. It has not been measured directly, and therefore the title could focus on 'First direct measurements of...'. However, I'm not sure that that would be a sufficiently substantial contribution. In the light of my general comment above I would focus the research and title on how these measurements improve model parameters and variables.
13	I don't believe "significant" has been demonstrated. I'm not sure "simultaneous" has either. The problem with the latter is that this can be taken to mean seconds or days; I think it would be better to quantify and to use poorly-defined adjectives.
16-17	Can this relationship between load change in distance be quantified?
17-19	This is a bit of a manifesto statement (which is also vague) and should be replaced with explicit findings.
48	Relative to what? Perhaps a reference be given to the surface albedo claim - or perhaps even to the inset panel of Figure 1.

54	Actually, I do not believe that this is the “ideal study site”. I would have thought that “ideal” study site would have one central surface lake that was deep surrounded by uniform homogeneous ice to a distant boundary.
Figure 1	Looking at the inset, it seems that at Peanut and Ring Location 3 should be the lake centre and Location 1 the margin, but I believe it is the other way around. I don’t dispute that this is the case but it is clear that these two locations 3 and 1 are located on different surface ice types, with 3 appearing as refrozen and uplifted lake ice or lake-bed ice. Does this likely difference in ice bulk density influence the results in any way?
78	“entrainment by”?
Figure 2	For me this Figure is not illustrating the interpreted pattern as clearly as it could. I think I would put panels a. to d. directly beneath each other and combine the three GPS records for each one into an upper sub-panel, differentiating between the three GPS records in each by line colour. I would say thickness, not I think a thickness band is needed for error. The y-axis label is “deviation (m)” while the caption states “GPS data...”. Together, these do not adequately describe what is being plotted. It appears to be ‘vertical deviation (or ‘surface elevation change’) since the start of the experiment’. Shaded error zones should be added to these GPS records. What does a bivariate plot of elevation (or change in elevation) against water depth (or change in water depth) - with each GPS plotted with a different colour code - look like? Would this not be a closer representation of the manuscript’s most important relationship (presented as well as, and not instead of, a time-series of the two independently)?
104 and elsewhere	I would encourage the manuscript to use m and mm, and avoid cm.
110 and 102	There’s a lot of “clear” being mentioned here. Why not the ‘most pronounced’? If the physics holds, why are the other responses ‘less clear or ‘unclear’? What is driving this variability in response? I imagine the answer lies in spatial variability in ice shelf internal structure (inherited stratification/foliation - Glasser), englacial water bodies (Roi Baudouin) and density (Larsen C) amongst others?
109	“22 December”
122-124	Second clause is redundant if the first clause holds.
127	There’s not much discussion here of the influence of the ice edge. Is this sufficiently far away the discounted from the analysis?

121	I would elevate this aspect of the work to about 50% of the paper. I think I would still present the field data in the way that they are, but reduce the discussion to basic data needed to calibrate the model. I will present more than the model's conceptual structure here.
142	What is the evidence for this claim of "extremely unlikely"?
137	This is interesting. Where is this mass lost to?
Figure 3	If the manuscript is restricted to presenting three figures, I think I would abandon this one as it addresses a contributory cause of the make-level fluctuations (while it is the level fluctuations themselves that cause the stress differentials that cause the flexure), and replace it with a plot of GPS-measured elevation against lake depth. If this figure remains, then volume errors should be plotted too.
162-163	Is this calculation correct?
169-170	Can this effect of insufficient flexure to generate crevasse is be evaluated in the model?
180-190	There is little either insightful or new in this paragraph. I think it is arguing that future melting is likely to lead to increased surface ponding, even though surface water can drain. As I read it, this entire paragraph could be in the introduction (I don't recommend it is included there, but I don't think it has any dependency on the results of this work). I would replace this paragraph with one that is more tightly aligned with the novel findings and interpretation of this work.
192-193	Again, "significant" and "simultaneous" need to be quantified in order for this statement to have value.
193-196	This final statement is almost identical to the rationale for the work presented in the introduction, and unfortunately pretty much holds unchanged. It leaves me pondering what the contribution of the manuscript actually has been in relation to addressing this rationale. I think the manuscript could go further in this regard and the final statement should really summarise that contribution. Indeed, the manuscript itself draws particular attention to this requirement in stating that "These results will be instrumental for improving ice-sheet models so that they are able to account for the flexural response of ice shelves...".
189	"affect"?

Reviewer #2 (Remarks to the Author):

This paper reports on field observations of ice-shelf flexure in response to supra-glacial lake drainage, which is a subject of considerable interest when considering ongoing and potential future climate response in Antarctica. The observations are novel, the methods to obtain them are both sensible and well described, and overall represent a substantial piece of work that will be appreciated in the wider ice shelf community.

I have a few minor comments, largely with the presentation rather than any scientific criticism.

L22: "...that weaken and flex.." → "... flex and can weaken" ?

L48: "surface melt rates are relatively high." could be quantified?

L85: "Meltwater, therefore, often pools to form active meltwater lakes in the low topographic areas around the relict lake scars" Are there depressed rings, relative to the far field, surrounding the relict lake pedestals?

L105 "[Ring] GPS 1 initially stays at a relatively constant elevation, before going down rapidly by ~30 cm day⁻¹ 106 on 21 and 22 December, corresponding to the time when the lake fills most rapidly, reaching > 2 m in depth." I agree that this feature looks to be present and related between the GPS and pressure curves, and the methods do make it clear the sampling rate is easily fine enough to resolve this, but there are features of similar magnitude (e.g in all three GPS at 05/01) that you don't try to explain until L143 "We also note that the short-lived reduction in the uplift rate around 5/6 144 January in all GPS datasets (Fig. 2) is likely due to a snow-fall event (Supplementary Movie 1)."

L122: "Compared to the GPS 1 station data at all lake sites, the data from GPSs 2 and 3 do not show a clear response to the filling/drainage of the lakes, even at the Ring site, where the largest GPS 1 uplift is measured in response to lake drainage (Fig. 2a)." Is that correct? To me, Ring seems unusual in that it is the only site that doesn't exhibit uplift in GPS 2 and 3. Rift tip in particular sees more than 0.5 m total uplift over all three GPS (which could be explained by a larger Young modulus, but also by a more widely distributed load, which you seem to suggest in L166).

L133: Does it make sense to report meltwater ponding/production volumes? I am guessing that you actually estimated the average depths over the discs and then multiplied by $2 \cdot \pi \cdot 250$.

L142 "this is extremely unlikely to be..." Why? Presumably it would require outlandish freezing rates (~ 100 m/a).

L168: There is a lot of 'may' in this paragraph, which might be reasonable, but if "lakes on 170 the McMIS simply do not currently reach large enough volumes to produce sufficient tensile-stress levels for fracture initiation" etc, seems to be at odds with the model which produces ~100 kPa stresses, and indeed you say "Alternatively, local flexure-induced tensile stresses may be sufficiently large for fracture" and refer to the model – that indicates that the volumes must be large enough, no?

Fig 1, caption: "A yellow star marks the location of the AWS". AWS has not been introduced at this point in the text. Maybe "The yellow star marks the location of an automatic weather station (AWS)"

Fig 2: Could the GPS 1/2/3 elevation plots for each site be placed on the same axes? E.g 2a would have one panel showing the three Ring GPS traces, 2b would have one panel showing the Rift Tip GPS, etc. the Ring and WT GPS it is clear that GPS 1 sees the greatest deflection, but it is not so clear for Rift Tip or Peanut. The text does say "At all lake sites during the melt season, the greatest changes in ice-shelf vertical elevation in response 90 to meltwater loading/unloading are close to the active lake centres", but I think it makes sense to have the plots show this clearly.

Supplementary Table 1. Does r (m) have the correct units (column 5)? Looking at supp. Fig 2a, the value should be 250m, but the table has 0.25 m. I.e should it be r (km) ?

Reviewer #3 (Remarks to the Author):

This paper uses in situ data of lake filling and drainage on an Antarctic ice shelf to demonstrate processes of ice flexure. Given the dramatic collapse of Larsen B from meltwater ponding and the subsequent focus on the role of ice flexure in this phenomenon, obtaining in situ data is a useful endeavor. The authors argue in their introduction that the flexure from ponding dynamics can cause crevasses to form within and on the outside of the basin. If they had captured the formation of the fractures that would have been excellent for application to flexure models and for understanding the role of meltponding in ice shelf stability. The first half of the paper reads as if the primary interest in the lakes is fracture formation. However, it then appears (only by line 120) that the lakes drained over the surface, not through fractures, and didn't cause any nearby fractures. As a result, the direction of the paper then shifts to discuss the net meltwater balance and the calculations thereof rather than application to flexure calculations and modeling. Because of this shift, it is hard to follow the primary aims and applications of this study.

I would like the authors to be more upfront about the fact that they didn't record hydrofracture and discuss why it didn't occur in more depth. Is it something to do with the velocity gradients? The ice thickness? Can they tell anything about the ice shelves that are more or less vulnerable to lake-related collapse from this study? Is there any evidence of other lakes on McMurdo hydrofracturing previously?

There is very little mentioned about the flexure model, which would appear to be the primary motivation behind the study i.e. to test in situ data against the models. The model discussion totals 6 lines (126-131). Instead much of the paper is dedicated to discussing melt rate calculations and I'm not sure what this adds to the aims of the paper as set out at the beginning i.e. the role of lakes in the stability of ice shelf lakes.

There is also the question whether a thin ice shelf like the McMurdo ice shelf which only has an ice thickness of 10-30m can translate up to the level of hundred + meter thick ice shelves like Larsen. Would the thinness of McMurdo have increased or decreased the likelihood of hydrofracture? The only GPS data that clearly show the loading unloading signal due to lake filling/drainage is Ring GPS1. Rift Tip GPS 1 shows a relationship for unloading (although not loading) but these data have been estimated and were not recorded by the GPS. I think given that the primary arguments of the authors are based on the GPS uplift rates it is not reasonable to present data (even as a dotted line) for this period of time when data are missing (Fig 2b). I have a similar argument against the estimated lake depth plots. For the other sites (and GPS) there seems to be little relationship

between the water depth and GPS.

Why does the elevation keep increasing for all GPS after the lakes have drained? The authors spend much of the paper discussing the net meltwater calculations and suggest loss of melt mass causes the uplift seen in all GPS following lake drainage. However I'm unclear how this would work. As the shelf is floating, the only way to increase the elevation should be to add mass onto the base. If mass is lost from the top, the shelf will rise slightly, which will offset that loss but shouldn't raise the GPS higher than they originally were. This is particularly problematic for this shelf which is so thin (estimated 10m from the flexure modeling) as a surface increase of 0.5m as seen in the weeks following lake drainage at the multiple sites is therefore a substantial increase in surface elevation for this ice shelf.

Specific comments

47 – how is the debris on the surface if it was transported by the LGM ice shelf?

48 – can you provide an estimate of how high the meltwater is here?

75 – what causes the topographically low areas? I see you discuss the pedestals from frozen lakes but what caused the topographic lows for those lakes to initially form in? Are all the topographic lows in the region related to pedestal former lakes?

133 – why did you calculate meltwater production within a 250m radius? And why do you do this for the 3 GPS rather than the catchment of the lake? I read the methods section but I'm still confused. Is the argument that the GPS are each impacted by the meltwater budget only in the 250m radius? Is there any flexure data to back that up – as all are raising at a similar rate would it not be a wider shelf melt rate that drives the uplift?

163-167- confusing (and long) sentence

180-184 – confusing sentence. If rivers are helping to evacuate water causing net meltwater budget I'm not sure more meltwater will encourage ponding and positive meltwater budgets - wouldn't it just make the rivers flow faster?

188 – potential fracture? I thought you didn't see any fractures?

191-196 – again the justification is given for application to models. Why not make more of the modeling that you did? That would seem the most useful application of this study.

356 – was the timelapse data used to validate the PDD calculation? What were the data used for?

395 – if the threshold was lowered that much does this mean other studies using the higher threshold are wrong? This could be an important point.

425 – given the debris content, how much error does this add to your calculations?

437 – what about the assumptions mentioned in lines 421-425?

441 – which ablation measurements? The ones in the lake? If measured against the GPS poles do you have multiple measurements or just between installation and retrieval?

438 – can you explain the PDD more fully to that it is clear how the >12 mm we/day becomes 47.5 mm we C-1 day⁻¹

492 – wouldn't the Maxwell time be in the range of several days? As your lakes take several weeks to drain it seems that a visco-elastic model would be more useful.

Fig1 – the lake extent is unclear from this (or any) figure so it's hard to understand the arguments about GPS 1 at each lake having the most uplift. This makes it very difficult to understand the GPS data in Figure 2 as it's not clear where these are located relative to the lake.

Fig2 – I don't see any downward deflection in the GPS apart from a small change in Ring GPS1 a few days before drainage. The overall trend for the GPS appear to be upward deflection. The WT and Peanut sites don't seem to have much of a break in slope apart from WT GPS 1. I had to read to the end of the paper before realizing some data in Figure 2a and b were estimated and not recorded – this should be clarified in the caption.

Surface meltwater movement, ponding and drainage causes ice-shelf flexure

The manuscript presents field-based data relating to the vertical surface movement of the Ross Ice Shelf at different distances from lake basins that fill and empty, demonstrating differential surface uplift and lowering. These data are also linked to a model of ice-shelf flexure. The data are hard-earned and deserve publication. However, that the ice shelf flexes in response to differential surface loading is not a particularly novel finding as it follows directly from physical considerations and has consequently been assumed by previous studies. Nonetheless, this manuscript does present the first direct measurements of such flexure (to my knowledge). However, these measurements are themselves not perfect, being driven by fairly small meltwater fluxes and potentially being influenced by the ice shelf margin. One particularly interesting aspect of the manuscript that is currently, in my opinion, underrepresented is the integration of these field data with a computer-based model of flexure response. The principal thrust of the rationale stated for the research is that these data are needed in order to contribute to models of ice shelf behaviour (lines 36 to 38 for example). As I see it, the manuscript potentially has this capacity but does not currently elevate the model structure or calibration and validation to the status that it should have to address this ambition. I would encourage the authors to recast the manuscript such that the field data are used explicitly to derive parameter and variable values for the model, allowing the model to be refined and presented in such a way that the process can be integrated into larger-scale models. In this way, I feel the novel value of the manuscript will be made clearer and the work will be more directly aligned to its rationale.

Some more specific comments follow:

Location/Line	Comment/Suggestion
Title	I think this could be written more specifically to reflect the novel advances of the research presented in this manuscript. That ponding and drainage causes flexure is not debated and has been assumed in the past. It has not been measured directly, and therefore the title could focus on 'First direct measurements of...'. However, I'm not sure that that would be a sufficiently substantial contribution. In the light of my general comment above I would focus the research and title on how these measurements improve model parameters and variables.
13	I don't believe "significant" has been demonstrated. I'm not sure "simultaneous" has either. The problem with the latter is that this can be taken to mean seconds or days; I think it would be better to quantify and to use poorly-defined adjectives.
16-17	Can this relationship between load change in distance be quantified?
17-19	This is a bit of a manifesto statement (which is also vague) and should be replaced with explicit findings.
48	Relative to what? Perhaps a reference be given to the surface albedo claim - or perhaps even to the inset panel of Figure 1.

54	Actually, I do not believe that this is the “ideal study site”. I would have thought that “ideal” study site would have one central surface lake that was deep surrounded by uniform homogeneous ice to a distant boundary.
Figure 1	Looking at the inset, it seems that at Peanut and Ring Location 3 should be the lake centre and Location 1 the margin, but I believe it is the other way around. I don’t dispute that this is the case but it is clear that these two locations 3 and 1 are located on different surface ice types, with 3 appearing as refrozen and uplifted lake ice or lake-bed ice. Does this likely difference in ice bulk density influence the results in any way?
78	“entrainment by”?
Figure 2	For me this Figure is not illustrating the interpreted pattern as clearly as it could. I think I would put panels a. to d. directly beneath each other and combine the three GPS records for each one into an upper sub-panel, differentiating between the three GPS records in each by line colour. I would say thickness, not I think a thickness band is needed for error. The y-axis label is “deviation (m)” while the caption states “GPS data...”. Together, these do not adequately describe what is being plotted. It appears to be ‘vertical deviation (or ‘surface elevation change’) since the start of the experiment’. Shaded error zones should be added to these GPS records. What does a bivariate plot of elevation (or change in elevation) against water depth (or change in water depth) - with each GPS plotted with a different colour code - look like? Would this not be a closer representation of the manuscript’s most important relationship (presented as well as, and not instead of, a time-series of the two independently)?
104 and elsewhere	I would encourage the manuscript to use m and mm, and avoid cm.
110 and 102	There’s a lot of “clear” being mentioned here. Why not the ‘most pronounced’? If the physics holds, why are the other responses ‘less clear or ‘unclear’? What is driving this variability in response? I imagine the answer lies in spatial variability in ice shelf internal structure (inherited stratification/foliation - Glasser), englacial water bodies (Roi Baudouin) and density (Larsen C) amongst others?
109	“22 December”
122-124	Second clause is redundant if the first clause holds.
127	There’s not much discussion here of the influence of the ice edge. Is this sufficiently far away the discounted from the analysis?

121	I would elevate this aspect of the work to about 50% of the paper. I think I would still present the field data in the way that they are, but reduce the discussion to basic data needed to calibrate the model. I will present more than the model's conceptual structure here.
142	What is the evidence for this claim of "extremely unlikely"?
137	This is interesting. Where is this mass lost to?
Figure 3	If the manuscript is restricted to presenting three figures, I think I would abandon this one as it addresses a contributory cause of the make-level fluctuations (while it is the level fluctuations themselves that cause the stress differentials that cause the flexure), and replace it with a plot of GPS-measured elevation against lake depth. If this figure remains, then volume errors should be plotted too.
162-163	Is this calculation correct?
169-170	Can this effect of insufficient flexure to generate crevasse is be evaluated in the model?
180-190	There is little either insightful or new in this paragraph. I think it is arguing that future melting is likely to lead to increased surface ponding, even though surface water can drain. As I read it, this entire paragraph could be in the introduction (I don't recommend it is included there, but I don't think it has any dependency on the results of this work). I would replace this paragraph with one that is more tightly aligned with the novel findings and interpretation of this work.
192-193	Again, "significant" and "simultaneous" need to be quantified in order for this statement to have value.
193-196	This final statement is almost identical to the rationale for the work presented in the introduction, and unfortunately pretty much holds unchanged. It leaves me pondering what the contribution of the manuscript actually has been in relation to addressing this rationale. I think the manuscript could go further in this regard and the final statement should really summarise that contribution. Indeed, the manuscript itself draws particular attention to this requirement in stating that "These results will be instrumental for improving ice-sheet models so that they are able to account for the flexural response of ice shelves...".
189	"affect"?

Response to reviews for “*Surface meltwater movement, ponding and drainage causes ice-shelf flexure*”, submitted to Nature Communications (new title: *Direct Measurements of Ice-Shelf Flexure caused by Surface Meltwater Ponding and Drainage*)

We are very grateful for the useful and constructive comments that the reviewers have provided. We are pleased to hear that they recognised our study is “novel”, that our data are “hard earned and deserve publication”, that we use “sensible and described” methods, and that our paper will be “appreciated by the wider ice shelf community”. As Reviewers 1 and 3 suggest that we should increase the length of the modelling component of the paper, we have responded to this important comment directly below, before responding to all other comments on the subsequent pages of this response letter.

The key aim of our study is to provide the *first direct measurements to show that changes in surface water loading does indeed cause ice shelf flexure and fracture*. This has been *suggested* in previous work, and *modelled* (including by us, Banwell and MacAyeal, in previous work), but no previous study has provided any direct field measurements to prove that this process occurs on ice shelves. This key aim is now described more clearly at the end of the introductory section (lines 46 - 51) and via the revised title of the manuscript.

In the current study we *do* use our field data to constrain key parameters in a *small-scale elastic flexure model*, and we run the model, to show: 1) that model calibration using our field data is possible, i.e. our optimal parameter values fall within sensible ranges of values (lines 152 – 156); and 2) that our field data make sense, i.e. given the changes in the water loads, the ice thickness, and other variables and boundary conditions, it makes sense that we only see vertical ice shelf deflection < 500 m of lake centres (lines 145 - 151). So our paper does include a substantial modelling component. In the previous version of the manuscript, the model that we use was mainly described in the Methods, and in the captions for Supplementary Fig. 3 and Supplementary Table. We now include more detail about this model in the revised manuscript’s main text, including the parameters that were constrained with our field data, the optimal values of those parameters, and the ranges that they were tested within (lines 152 – 156). We have also moved the plot of the model ‘best results’ from the Supplementary Information (then Supp. Fig. 3) into the main paper (now Fig. 3). (i.e. the plot of the model results).

With the changes described above now made, we hope we have managed to better clarify the main aims of our paper, which focus on the first direct measurements of ice shelf flexure induced by changing water loading, and the first use of such data to constrain a small-scale elastic flexure model. Ultimately, our intention is that the field data and modelling results from this study will be useful for developing and constraining large *continent-scale ice-sheet models*, e.g. DeConto and Pollard (2016), and this is now clarified on lines 46 - 51. This is quite a big undertaking and, we hope the referees agree, is beyond the scope of the present paper. We are already in discussion with Rob DeConto about such a follow-up model development/calibration study, which would be aimed at a slightly different, more specialist audience compared to the broad readership that we hope our current study will be of interest to.

Please note that our responses to the three reviewer’s comments (below) are in blue, and line numbers refer to our revised manuscript *without track changes* (though we have also included a version of our revised manuscript *with track changes* for review).

Reviewer 1

The manuscript presents field-based data relating to the vertical surface movement of the Ross Ice Shelf at different distances from lake basins that fill and empty, demonstrating differential surface uplift and lowering. These data are also linked to a model of ice-shelf flexure. The data are hard-earned and deserve publication. However, that the ice shelf flexes in response to differential surface loading is not a particularly novel finding as it follows directly from physical considerations and has consequently been assumed by previous studies. Nonetheless, this manuscript does present the first direct measurements of such flexure (to my knowledge).

We accept the reviewer's point that our finding that "*the ice shelf flexes in response to differential surface loading ... follows directly from physical considerations and has consequently been assumed by previous studies*". However, as also highlighted by this reviewer, the important point is that our manuscript presents "***the first direct measurements of such flexure***". We have edited the title of our paper to emphasize that this is the key finding. The title is now: "Direct Measurements of Ice-Shelf Flexure caused by Surface Meltwater Ponding and Drainage".

However, these measurements are themselves not perfect, being driven by fairly small meltwater fluxes and potentially being influenced by the ice shelf margin.

Field measurements will never "*perfect*", as they might be in a laboratory setting, for example. Given the difficulties of finding depressions on the McMIS due to its relatively flat topography and lack of accurate satellite-derived DEMs prior to our fieldwork, and to the general difficulties of travelling around the "rough" surface of the ice shelf with hidden pockets of water, we think we did well to instrument the ice shelf as we did.

Contrary to the referee's statement about melt water fluxes, we argue that our melt rates are relatively high compared to other ice shelves due to the relatively low albedo surface of the McMIS (due to the large quantities of debris). Please see lines 65 - 67 *where we now state the mean (and range) of the ablation rates measured in our study area*. That combined with the thin ice in our study site (30 m), means that our measured flexure rates and magnitudes are relatively high.

Finally, we have no field evidence that the flexure we measure is influenced by the ice shelf margin. Our modelling results agree with this, on the basis that the closest lake centre is at least 10 flexural length scales from the ice front. The data we present in the paper *do* suggest, however, that surface lowering (uplift) is influenced by meltwater ponding (draining).

One particularly interesting aspect of the manuscript that is currently, in my opinion, underrepresented is the integration of these field data with a computer-based model of flexure response. The principal thrust of the rationale stated for the research is that these data are needed in order to contribute to models of ice shelf behaviour (lines 36 to 38 for example). As I see it, the manuscript potentially has this capacity but does not currently elevate the model structure or calibration and validation to the status that it should have to address this ambition. I would encourage the authors to recast the manuscript such that the field data are used explicitly to derive parameter and variable values for the model, allowing the model to be refined and presented in such a way that the process can be integrated into larger- scale models. In this way, I feel the novel value of the manuscript will be made clearer and the work will be more directly aligned to its rationale.

For our response to this point, please refer to our general comment on page 1 of this document. As explained, our paper does integrate the field data with our small-scale flexure model, and this aspect of the paper has now been clarified and elevated from the Methods and Supplementary Material sections, where it previously was into the main body of the paper. This included moving Supp. Fig. 3 (i.e. the model results) into the main paper (now Fig. 3). In the revised manuscript, we also now state the calibrated values for the key parameters that our simple elastic model finds in order to best match the measured data (please see lines 152 – 156).

Some more specific comments follow:

Title - I think this could be written more specifically to reflect the novel advances of the research presented in this manuscript. That ponding and drainage causes flexure is not debated and has been assumed in the past. It has not been measured directly, and therefore the title could focus on ‘First direct measurements of...’. However, I’m not sure that that would be a sufficiently substantial contribution. In the light of my general comment above I would focus the research and title on how these measurements improve model parameters and variables.

We thank the reviewer for this suggestion, and have changed the title to “*Direct Measurements of Ice-Shelf Flexure caused by Surface Meltwater Ponding and Drainage*”. Although, as explained previously, we successfully calibrate key parameters in our small-scale elastic model using our field data (see lines 152 – 156 and Fig. 3), we think including this in the papers title would make it rather unwieldy and would prefer to stick with the revised title, focussing on the field data, if you agree.

13 - I don’t believe “significant” has been demonstrated. I’m not sure “simultaneous” has either. The problem with the latter is that this can be taken to mean seconds or days; I think it would be better to quantify and to use poorly-defined adjectives.

As the McMIS in our field area is only 10 – 30 m thick in our study area, and as we measure vertical ice uplift of up to 1 m over a time period of 5 weeks, we think the uplift measured could be described as ‘significant’. However, we have replaced ‘significant’ with ‘pronounced’, which is word that this reviewer suggests later in their review. Likewise, given the ice shelf began to uplift within two hours of Ring Lake starting to drain (Figure 2), we think it could be described as ‘simultaneous’. However, we have replaced ‘simultaneous’ with ‘immediate’ on all occasions.

16-17 - Can this relationship between load change in distance be quantified?

Yes, this can be quantified. We have reworded this sentence to say that the magnitude of the vertical ice-shelf deflection “decreases from maxima of ~1 m at the centres of the maximum load changes to zero at distances of ≤ 500 m” (lines 17 – 19).

17-19 - This is a bit of a manifesto statement (which is also vague) and should be replaced with explicit findings.

We have rewritten this sentence so that it now also mentions the process-scale model that we constrain in the *current* study: “Our results are used to constrain key parameters in a process-scale model of ice-shelf flexure, and will ultimately help improve the physical representation of ice shelves in current continent-wide ice-sheet models.” (lines 20 - 22). We would like to include this statement, if you agree, as it describes an important potential use of the field data and small-scale model results that we present in this manuscript.

48 - Relative to what? Perhaps a reference be given to the surface albedo claim - or perhaps even to the inset panel of Figure 1.

We mean that, due to the low albedo of the debris on the McMIS, the ablation rates are higher than they are on other ice surfaces at the same latitude. This is now clarified in the text (lines 60 - 61), and we also now state the average ablation rate that we measured (line 66).

54 - Actually, I do not believe that this is the “ideal study site”. I would have thought that “ideal” study site would have one central surface lake that was deep surrounded by uniform homogeneous ice to a distant boundary.

On the basis that the lakes we studied were not axisymmetric, with homogeneously deep water, surrounded by uniform and homogeneously thick ice, we agree with this reviewer that our study sites was not “ideal”. We have therefore we have replaced “ideal” with “practical”. However, we also note that although uniform/homogenous conditions are often assumed for experiments in labs, or in theoretical models, such conditions almost never exist in the real world. We were pleased at how sufficiently good our lake sites proved to be.

Figure 1 - Looking at the inset, it seems that at Peanut and Ring Location 3 should be the lake centre and Location 1 the margin, but I believe it is the other way around. I don't dispute that this is the case but it is clear that these two locations 3 and 1 are located on different surface ice types, with 3 appearing as refrozen and uplifted lake ice or lake-bed ice. Does this likely difference in ice bulk density influence the results in any way?

“Active lakes” (i.e. those that currently fill and drain in the melt season) formed in the dirty, low elevation areas around the uplifted lake ice (i.e. which we call “lake scars”). GPS 1s are in the centre of the active lakes, GPS 3s are in the centre of the uplifted relict lake scars. Please see lines 90 – 103 for clarification. The bulk density does vary spatially, but we have little evidence that it changes in response to debris movement. Also, any bulk density differences due to debris have been in place for 10's to 1000's of years (Kellog et al 1991), and thus do not drive the short term ice-shelf flexure effects that we measured.

78 - “entrainment by”?
Changed accordingly.

Figure 2 - For me this Figure is not illustrating the interpreted pattern as clearly as it could. I think I would put panels a. to d. directly beneath each other and combine the three GPS records for each one into an upper sub-panel, differentiating between the three GPS records in each by line colour. I would say thickness, not I think a thickness band is needed for error. After having experimented with the reviewer's suggestion, we do not think that doing this would make the figure clearer. This is because many of the data lines would end up being superimposed on one another, and thus would obscure one another. For example, in the Ring Lake data, the three GPS readings are virtually the same until the lake drainage on around 21/22 December, when GPS 1 (only) starts to rise rapidly.

Regarding errors on the GPS data, we have now added a few sentences to describe these in the GPS processing section of the Methods (lines 372 - 276). However, we do not think it is necessary to add error bands to the plots (i.e. Fig 2), and this is not commonly done in the glaciological literature (e.g. Bartholomew et al 2010, Tedesco et al 2013; Das et al 2008 and many other studies that plot vertical elevation data derived from GPS data do not include error bars). Although it may seem odd that error bars are not commonly added, this is because GPS position error is a complex function of time that becomes vanishingly small (in terms of sensitivity, not perhaps in terms of absolute location) as time-averaging of data is applied.

The y-axis label is “deviation (m)” while the caption states “GPS data...”. Together, these do not adequately describe what is being plotted. It appears to be ‘vertical deviation (or ‘surface elevation change’) since the start of the experiment’.

The y-axis labels are actually just “elevation (m)”. However, we thank the reviewer for pointing out that the use of ‘GPS data’ in the caption is not the best choice of words. We have now edited the caption to: “Vertical ice-shelf elevation data from the 12 GPS stations in our study area...”. We also now state the following bit of information in the caption for Figure 2: “All vertical elevation time series start at 0 m on 13 November”.

Shaded error zones should be added to these GPS records.
Please see our comment above about error bars/zones.

What does a bivariate plot of elevation (or change in elevation) against water depth (or change in water depth) - with each GPS plotted with a different colour code - look like? Would this not be a closer representation of the manuscript’s most important relationship (presented as well as, and not instead of, a time-series of the two independently)?

We thank this reviewer for their suggestion, but we do not think that this plot will be useful as it is *changes in water volumes* (and therefore loading) near to a GPS, that should be correlative with the vertical elevation changes measured by the GPS, *not* changes in water depth at a single point. It is for this reason that we also calculated the net meltwater budget (i.e. a volume) through time around each GPS site, and compared those plots with the GPS data.

104 and elsewhere - I would encourage the manuscript to use m and mm, and avoid cm.
Good point. We now only use mm and m as units throughout the manuscript.

110 and 102 - There’s a lot of “clear” being mentioned here. Why not the ‘most pronounced’? If the physics holds, why are the other responses ‘less clear or ‘unclear’? What is driving this variability in response? I imagine the answer lies in spatial variability in ice shelf internal structure (inherited stratification/foliation - Glasser), englacial water bodies (Roi Baudouin) and density (Larsen C) amongst others?

Thank you - we agree that ‘most pronounced’ is better wording than ‘clear’, and have made this edit. The variability in the responses between the various GPSs at the 4 lakes sites is due mainly to the change in total water volume (and hence loading/unloading) in the vicinity of each site, and rate at which those loading/unloading changes happen. As this reviewer suggests, the spatially variable ice shelf response may also be due to the spatially variable internal structure of the ice, but we think that this will be much less of a controlling factor compared with the changes in water load magnitudes, especially as the 4 sites are relatively close to one another (< 5 km) and we know that the ice has a fairly uniform thickness here (e.g. Rack et al 2013; Campbell et al., 2017).

109 - “22 December”
Changed

122-124 - Second clause is redundant if the first clause holds.
We have deleted the second part of this sentence, which we agree was unnecessary.

127 - There’s not much discussion here of the influence of the ice edge. Is this sufficiently far away the discounted from the analysis?

Yes, as the ice front is at least 5 km from Ring Lake centre (which we run the model for), our results show that ice-front effects can be discounted from our analysis (i.e. as the distance from the Ring lake's centre to the ice front is about 20 x the flexural length scale). We have also now added a sentence in the modelling section of our Methods to state that effects from the ice front can be discounted: "An undeflected ice shelf at a large distance (i.e. $r \leq \infty$) is assumed, which is appropriate because the grounding lines and ice front are > 2 km from all lake centres" (lines 537 – 539)

121 - I would elevate this aspect of the work to about 50% of the paper. I think I would still present the field data in the way that they are, but reduce the discussion to basic data needed to calibrate the model. I will present more than the model's conceptual structure here. Please see our earlier main comment in response to this (page 1).

142 - What is the evidence for this claim of "extremely unlikely"?

It is unlikely (we have now removed the word 'extremely' to make the statement less strong) that sub ice-shelf accumulation is the main control on the vertical uplift trend observed at all GPS stations during the melt season because sub-ice-shelf accumulation is balanced by surface ablation on an annual basis (Glasser et al, 2006). Therefore, as surface ablation is only prevalent in the summer, but sub ice-shelf accumulation happens all year round, surface ablation *must* be the dominant control on ice shelf uplift in the summer. This is now more clearly explained on lines 177 - 180.

137 - This is interesting. Where is this mass lost to?

It is either lost to other areas of the ice shelf, or more likely, to the ocean. We have now added the following statement to the manuscript: "The mass, i.e. meltwater, is being lost either to other areas of the ice shelf or, more likely, to the ocean transported via surface streams" (lines 175 - 177).

Figure 3 - If the manuscript is restricted to presenting three figures, I think I would abandon this one as it addresses a contributory cause of the lake-level fluctuations (while it is the level fluctuations themselves that cause the stress differentials that cause the flexure), and replace it with a plot of GPS-measured elevation against lake depth.

If this figure remains, then volume errors should be plotted too.

We are restricted to 4 figures, and currently have 3 in the main text. We have chosen to keep this figure as we think it is very useful in explaining, for example, why GPS 1 rises more than the other two GPS for this lake site (with the same trend occurring for the other 3 lake sites – see Supplementary Fig 4). The figure also shows why GPSs 2 and 3 are also rising in elevation during the melt season. The reviewer suggests that we instead insert a figure showing GPS-measured elevations plotted alongside lake measures depths for all GPS stations, but these are already shown in Figure 2.

We agree that it is important to plot errors where appropriate, however in this instance the errors are negligible and it is not appropriate to plot them. The reasons for why they are negligible is explained on lines 489-492 in the Methods.

162-163 - Is this calculation correct?

Yes, $5.7 \times 10^4 \text{ m}^3$ is correct, but the value for the max. meltwater budget (earlier in the sentence) was wrong and we apologise for that error. It has now been corrected (line 200). We have also now double-checked all other calculations, and they are correct.

169-170 - Can this effect of insufficient flexure to generate crevasse be evaluated in the model?

Unfortunately it cannot, as we do not know what the stress criterion for fracture is for this ice shelf. Although we know the ice thickness, the fracture criterion also depends on many other factors, including: i) background stress regime, which is more dependent on embayment geometry and ice stream input than ice thickness (Rignot et al, 2011); ii) depth-varying ice temperature (MacAyeal and Thomas, 1985); iii) Presence of prior damage (i.e. pre-existing crevasses) (Borstad et al 2013); and iv) the timescales of loading and unloading of surface meltwater, which determine whether an ice shelf behaves elastically or viscoelastically (MacAyeal and Sergienko, 2013; Banwell and MacAyeal, 2015). For these reasons, fracture thresholds and collapse vulnerability linked directly to ice shelves thicknesses are difficult to constrain, although this has been attempted (Albrecht and Levermann, 2012).

180-190 - There is little either insightful or new in this paragraph. I think it is arguing that future melting is likely to lead to increased surface ponding, even though surface water can drain. As I read it, this entire paragraph could be in the introduction (I don't recommend it is included there, but I don't think it has any dependency on the results of this work). I would replace this paragraph with one that is more tightly aligned with the novel findings and interpretation of this work.

We now start this paragraph with one of the key results of our study, i.e. that “our results indicate that there is currently a negative net meltwater budget on the McMIS during the melt season...”, and we then discuss the significance of this result in the context of other studies. As this paragraph (the penultimate one of the paper) is part of the Discussion/Conclusion section, we think it is useful to consider our study's results in the context of the eight other studies that we mention and reference here.

192-193 - Again, “significant” and “simultaneous” need to be quantified in order for this statement to have value.

We have changed these words. Please refer to our earlier response regarding this comment (original manuscript line 13).

193-196 - This final statement is almost identical to the rationale for the work presented in the introduction, and unfortunately pretty much holds unchanged. It leaves me pondering what the contribution of the manuscript actually has been in relation to addressing this rationale. I think the manuscript could go further in this regard and the final statement should really summarise that contribution. Indeed, the manuscript itself draws particular attention to this requirement in stating that “These results will be instrumental for improving ice-sheet models so that they are able to account for the flexural response of ice shelves...”

Please see our earlier general comment regarding this on page 1. Additionally, in this conclusion section, we do now state the results of our current study, including a comment about how our field observations compare with our process-scale model.

189 - “affect”?

Changed.

Reviewer 2

This paper reports on field observations of ice-shelf flexure in response to supra-glacial lake drainage, which is a subject of considerable interest when considering ongoing and potential

future climate response in Antarctica. The observations are novel, the methods to obtain them are both sensible and well described, and overall represent a substantial piece of work that will be appreciated in the wider ice shelf community.

We thank this reviewer for their comments and are pleased that they think we chose suitable field methods, have novel results, and have produced a well-written manuscript that will be a useful addition to the current ice shelf literature.

I have a few minor comments, largely with the presentation rather than any scientific criticism.

L22: "...that weaken and flex.." → "... flex and can weaken" ?
Changed accordingly.

L48: "surface melt rates are relatively high." could be quantified?

This comment is very similar to one of reviewer 1's comments, and we have reworded this sentence to say: "This gives the surface a low albedo, which means that although the McMIS is relatively far south (~77°), surface melt rates are relatively high compared to other ice masses at similar latitudes" (lines 59 - 61). We have also now stated the mean melt rate (and range) that we measured from 12 GPS stakes (lines 65 - 67).

L85: "Meltwater, therefore, often pools to form active meltwater lakes in the low topographic areas around the relict lake scars" Are there depressed rings, relative to the far field, surrounding the relict lake pedestals?

These areas are not 'rings' in the sense that they were formed due to the drainage and viscoelastic rebound of a central, pedastaled lake. These areas (which often look ring-like in the sense that they surround a higher, frozen-over lake) are simply where water collects due to the relatively low elevation of these areas. Please see lines 99 – 103.

L105 "[Ring] GPS 1 initially stays at a relatively constant elevation, before going down rapidly by ~30 cm day⁻¹ on 21 and 22 December, corresponding to the time when the lake fills most rapidly, reaching > 2 m in depth." I agree that this feature looks to be present and related between the GPS and pressure curves, and the methods do make it clear the sampling rate is easily fine enough to resolve this, but there are features of similar magnitude (e.g in all three GPS at 05/01) that you don't try to explain until L143 "We also note that the short-lived reduction in the uplift rate around 5/6 144 January in all GPS datasets (Fig. 2) is likely due to a snow-fall event (Supplementary Movie 1)."

We delay describing the effect of the snowfall event on 5/6 January until later in the manuscript as meltwater ponding/drainage-induced flexure is the main focus of this study. The reduction of the ice uplift rate due to the snowfall event is simply an interesting observation that we decided to mention briefly. Although the initial depression of the ice shelf on 21/22 Dec is similar in magnitude to the snow fall event on 5/6 Jan (both about 20 cm), the depression on 21/22 Dec is immediately followed by rapid uplift of the ice shelf (by about a metre), whereas the depression of the ice shelf on 5/6 Jan is not followed by such uplift. This suggests that the ice shelf depression on 5/6 is *not* in response to the filling/draining of a lake, unlike the depression on 21/22 Dec. Further, we can see from the movie produced from the time-lapse camera (Supp. Movie 1) that we have interpreted the causes of these two depression events correctly. This is now clarified on line 182.

L122: "Compared to the GPS 1 station data at all lake sites, the data from GPSs 2 and 3 do not show a clear response to the filling/drainage of the lakes, even at the Ring site, where the

largest GPS 1 uplift is measured in response to lake drainage (Fig. 2a).” Is that correct? To me, Ring seems unusual in that it is the only site that doesn’t exhibit uplift in GPS 2 and 3. Rift tip in particular sees more than 0.5 m total uplift over all three GPS (which could be explained by a larger Young modulus, but also by a more widely distributed load, which you seem to suggest in L166).

We suggest that this reviewer has not quite understood the sentence that they are referring to, which is our fault, as it was badly worded. Following reviewer 1’s suggestion, we have now re-worded it to say: “Compared to the GPS 1 station data at all lake sites, no data from GPSs 2 and 3 show a pronounced response to the filling/drainage of the lakes (Fig. 2a).” Yes all GPS stations do show an upward trend, particularly late in the season when lakes have drained, which is due to a generally negative net meltwater budget, however this sentence is specifically evaluating GPS response to lake drainage, which is only pronounced at the GPS 1s (i.e. nearest to the maximum change in load at each lake site).

L133: Does it make sense to report meltwater ponding/production volumes? I am guessing that you actually estimated the average depths over the discs and then multiplied by $2\pi \times 250$.

Yes we think it *does* make sense to report meltwater ponding/production volumes, but we did *not* calculate these by estimating the water depths and meltwater production rates over the circles and then multiplying them by $2\pi \times 250$. The way we calculate meltwater ponding and production is described in our Methods. For the *calculations of volumes of meltwater ponding*, we analysed *all* pixels in each circle to determine whether they were water covered or not, and if they were, the depth of water on each pixel was calculated (see “Calculating volumes of observed surface meltwater ponding” section in the Methods for more information). For the *calculations of volumes of meltwater production*, we calculated two PDD factors, one for clean ice, and one for dirty ice, and then apportioned all pixels within each circle as either “clean” or “dirty” ice (see “Calculating volumes of surface meltwater production” section in the Methods for more information).

L142 “this is extremely unlikely to be...” Why? Presumably it would require outlandish freezing rates (~ 100 m/a).

Please see our answer to Reviewer 1’s similar comment regarding original lines of this response letter.

L168: There is a lot of ‘may’ in this paragraph, which might be reasonable, but if “lakes on the McMIS simply do not currently reach large enough volumes to produce sufficient tensile-stress levels for fracture initiation” etc, seems to be at odds with the model which produces ~ 100 kPa stresses, and indeed you say “Alternatively, local flexure-induced tensile stresses may be sufficiently large for fracture” and refer to the model – that indicates that the volumes must be large enough, no?

As we explain above in response to a comment by Reviewer 1 (for original manuscript lines 169 – 170), we do not (and cannot) know what the stress criterion for fracture for this area of the McMIS. Therefore, we state in the current manuscript: This may be because: hydrofracture does not assist flexure-driven fracture when ice-shelf thicknesses are small, because the hydrostatic head can never become large enough³⁰; and/or lakes on the McMIS simply do not currently reach large enough volumes to produce sufficient tensile-stress levels for fracture initiation (lines 207 – 211). Please see lines 211 – 219 for further explanation.

Fig 1, caption: “A yellow star marks the location of the AWS”. AWS has not been introduced

at this point in the text. Maybe “The yellow star marks the location of an automatic weather station (AWS)”

Changed.

Fig 2: Could the GPS 1/2/3 elevation plots for each site be placed on the same axes? E.g 2a would have one panel showing the three Ring GPS traces, 2b would have one panel showing the Rift Tip GPS, etc. the Ring and WT GPS it is clear that GPS 1 sees the greatest deflection, but it is not so clear for Rift Tip or Peanut. The text does say “At all lake sites during the melt season, the greatest changes in ice-shelf vertical elevation in response 90 to meltwater loading/unloading are close to the active lake centres”, but I think it makes sense to have the plots show this clearly.

Please see our response to Reviewer 1’s comment, who also suggests we put the GPS 1, 2 and 3 data for each site on the same axis. Briefly, we have chosen not to follow this suggestion as in the early season, the lines will be superimposed on top of one another.

Supplementary Table 1. Does r (m) have the correct units (column 5)? Looking at supp. Fig 2a, that value should be 250m, but the table has 0.25 m. I.e should it be r(km) ?

Thank you for noticing this error; the units are now correct.

Reviewer #3 (Remarks to the Author):

This paper uses in situ data of lake filling and drainage on an Antarctic ice shelf to demonstrate processes of ice flexure. Given the dramatic collapse of Larsen B from meltwater ponding and the subsequent focus on the role of ice flexure in this phenomenon, obtaining in situ data is a useful endeavor.

The authors argue in their introduction that the flexure from ponding dynamics can cause crevasses to form within and on the outside of the basin. If they had captured the formation of the fractures that would have been excellent for application to flexure models and for understanding the role of melt ponding in ice shelf stability. The first half of the paper reads as if the primary interest in the lakes is fracture formation. However, it then appears (only by line 120) that the lakes drained over the surface, not through fractures, and didn’t cause any nearby fractures. As a result, the direction of the paper then shifts to discuss the net meltwater balance and the calculations thereof rather than application to flexure calculations and modeling. Because of this shift, it is hard to follow the primary aims and applications of this study.

The reviewer has made a very useful point here. We have reworded sections of the Introduction so that there is less of a shift in the direction of the paper, and *more of an overall focus on the process of flexure*, which is a precursor to fracture. Although the Introduction does mention fracture, the focus of this section soon turns to flexure. For example, one sentence has been reworded to say: “Consequently, the process of ice-shelf flexure in response to surface meltwater load changes⁹, which may lead to fracture⁷, has yet to be incorporated into larger-scale ice-sheet models”. We also now clearly state that fractures in continent-scale ice sheet “models are not able to form as a result of ice-shelf flexure” (lines 44 -45). We also note that although we observed meltwater loading-induced flexure, we did not observe fracture (for the reasons stated on lines 207-219).

I would like the authors to be more upfront about the fact that they didn’t record hydrofracture and discuss why it didn’t occur in more depth. Is it something to do with the velocity gradients? The ice thickness? Can they tell anything about the ice shelves that are

more or less vulnerable to lake-related collapse from this study? Is there any evidence of other lakes on McMurdo hydrofracturing previously?

As we explain on lines 207 - 219, we do not know for sure why we did not observe hydrofracture, however we now give two possible explanations in the manuscript: i) hydrofracture may not assist flexure-driven fracture when ice-shelf thicknesses are small (i.e. only 10 – 30 m in our study area) as the hydraulic head may not be able to reach the sufficient depth (e.g. Van der Veen, 1998); or ii) lakes on the McMIS simply do not currently reach large enough volumes to produce sufficient tensile-stress levels for fracture initiation. As mentioned in our response to Reviewer 1's similar comment (original manuscript lines 169 - 170), it is impossible to know a priori the stress criterion for fracture for the McMIS. This is because although we know the ice thickness, the stress criterion for fracture also depends on many other factors, including: i) background stress regime, ii) depth-varying ice temperature iii) presence of prior damage; and iv) the timescales of loading/unloading of surface meltwater. For these reasons, fracture thresholds and collapse vulnerability linked directly to ice shelves thicknesses are difficult to constrain, although this has been attempted (Albrecht and Levermann, 2012). There is no evidence of other lakes on the McMIS having drained by hydrofracture previously, but that doesn't mean to say that it has not happened.

There is very little mentioned about the flexure model, which would appear to be the primary motivation behind the study i.e. to test in situ data against the models. The model discussion totals 6 lines (126-131). Instead much of the paper is dedicated to discussing melt rate calculations and I'm not sure what this adds to the aims of the paper as set out at the beginning i.e. the role of lakes in the stability of ice shelf lakes.

The primary motivation behind the study, which we now stress more clearly at the outset (lines 46 – 49) is to provide the *first direct field measurements of ice shelf flexure in response to surface lake filling and drainage*. A secondary aim is to use those in situ data to constrain a process-scale scale model (line 50). As explained in our longer comment about this on page 1 of this response letter, we have now increased the length of the small-scale modelling section by stating the key parameter values that we have constrained with our field data in the main body of the paper (lines 152-156) as well as bringing the previous Supplementary Fig 3 into the main body of the paper (new Figure 3). We also show more model results for a range of parameter values in Supplementary Table 1). In addition to showing that such a dataset can be used to constrain flexure models, the small scale modelling that we do, along with the meltwater budget calculations, are to help explain and understand the measured vertical ice shelf deflection at each GPS.

There is also the question whether a thin ice shelf like the McMurdo ice shelf which only has an ice thickness of 10-30m can translate up to the level of hundred + meter thick ice shelves like Larsen. Would the thinness of McMurdo have increased or decreased the likelihood of hydrofracture?

As explained in our response above, the stress criterion for fracture of an ice shelf depends on many factors in addition to simply ice thickness. Therefore, just because the McMIS is much thinner than the Larsen C, for example, this in itself neither increases nor decreases its chance of fracture in response to meltwater loading. We also suggest in the paper (line 208) that the ice in our study area may simply be too thin for hydrofracture to occur, as the hydraulic head in a fracture may never be large enough.

The only GPS data that clearly show the loading unloading signal due to lake filling/draining is Ring GPS1. Rift Tip GPS 1 shows a relationship for unloading (although not loading) but these data have been estimated and were not recorded by the GPS. I think given that the

primary arguments of the authors are based on the GPS uplift rates it is not reasonable to present data (even as a dotted line) for this period of time when data are missing (Fig 2b). I have a similar argument against the estimated lake depth plots. For the other sites (and GPS) there seems to be little relationship between the water depth and GPS.

Yes, the straight dashed lined in the Rift Tip GPS 1 plot shows estimated data, but these data are constrained at the beginning and end of the data period by measured data. This is now explained in the figure caption (lines 117 - 118). Therefore, although we think it is sufficiently useful to keep this dashed line, we are happy to remove it if required. But our preference is to keep it.

The two dashed water depth plots (for the loose sensors at Ring and Rift GPSs 1) are also estimated, but again, their start and end points are based on measurements (i.e. we know their start depth, as they were installed next to the 'fixed' sensors, and we know their end depth because we measured the total water depth when they were retrieved. The intervening data assume a constant lake bottom ablation rate during their deployment. This is now explained in the caption (lines 118-120). However, again, if you would prefer us to remove these dashed lines, we can although our preference is to keep them.

Why does the elevation keep increasing for all GPS after the lakes have drained? The authors spend much of the paper discussing the net meltwater calculations and suggest loss of melt mass causes the uplift seen in all GPS following lake drainage. However I'm unclear how this would work. As the shelf is floating, the only way to increase the elevation should be to add mass onto the base. If mass is lost from the top, the shelf will rise slightly, which will offset that loss but shouldn't raise the GPS higher than they originally were. This is particularly problematic for this shelf which is so thin (estimated 10m from the flexure modeling) as a surface increase of 0.5 m as seen in the weeks following lake drainage at the multiple sites is therefore a substantial increase in surface elevation for this ice shelf.

Here we think that this reviewer may have misunderstood the concepts of buoyancy, or forgotten that our GPS poles are fixed into the ice at depth. If the ice shelf is floating, it will rise upward (to remain in hydrostatic equilibrium) if mass is added to its underside, but it will *also rise upwards if it loses mass from its surface* (again to remain in hydrostatic equilibrium). Imagine a boat floating on water, with an elephant in it (!) If the elephant gets out, the boat will immediately move upwards and will float higher in the water.

It is also important to point out here that the GPSs are attached to poles that are drilled into the ice. Therefore they move vertically when the ice shelf does. Different behaviour would be observed if the GPSs simply rested on the melting ice shelf surface. The fact that the GPSs are drilled into the ice is clarified on line 72.

Specific comments

47 – how is the debris on the surface if it was transported by the LGM ice shelf?

Debris is exposed in the ice shelf surface due to high surface ablation rates that are balanced by sub-ice-shelf accumulation (Glasser et al 2016). Therefore, debris effectively moves upwards in the ice shelf. We have now added a few extra words to say this (lines 58-59).

48 – can you provide an estimate of how high the melt rates are here?

We have now added a sentence that says: “... we measured a mean surface melt rate of 5.2 mm w.e. day⁻¹ against 12 ablation stakes (range = 1.0 – 20.6 mm) from early Nov. 2016 to late Jan. 2017” (lines 65 - 67).

75 – what causes the topographically low areas? I see you discuss the pedastals from frozen

lakes but what caused the topographic lows for those lakes to initially form in? Are all the topographic lows in the region related to pedestal former lakes?

An ice shelf will naturally have undulations due to variability in ice flow, ice density and basal crevassing, differential ablation and accumulation etc. Over time, as water collects in topographic lows, they may become even lower due to reduced albedo of ponded water and enhanced ablation. On the McMIS, which has a variable debris cover, debris collecting in the topographic lows may also lower albedo and enhance ablation. We think that all the pedestaled lakes would have originally been lakes in the low dirtier areas, but which became pedestaled once they froze over, incorporating snow, giving them a high albedo surface, thereby reducing their ablation rate. Explaining the detailed formation of these lakes is beyond the scope of this paper, however, we have another paper in preparation about the evolution of these lakes on the McMIS, which uses Landsat and Sentinel 1 imagery from the last 10 years.

133 – why did you calculate meltwater production within a 250m radius? And why do you do this for the 3 GPS rather than the catchment of the lake? I read the methods section but I'm still confused. Is the argument that the GPS are each impacted by the meltwater budget only in the 250m radius? Is there any flexure data to back that up – as all are raising at a similar rate would it not be a wider shelf melt rate that drives the uplift?

We analyse the net meltwater budget in circles of a set radius around each GPS because we were interested in explaining the measured vertical deflection of each GPS in response to the loading/unloading of meltwater. As such, it only made sense to analyse *meltwater production and ponding* on ALL pixels in a circular area around each GPS. It would not have made sense to analyse only the pixels in an irregular shaped area around each GPS (e.g. a lake catchment, which would have been exceedingly hard/impossible to define due to the large scale flat nature of the ice shelf surface with its complicated small scale topography), as meltwater will also be being produced, and may be ponding in small quantities, on the intervening pixels, which will also effect the GPS loading/unloading.

Regarding the reasons for our choice of circle size, these are explained in the Methods (lines 420 - 425), but the key thing is that changing the radii of the circles did not change the main meltwater budget trends. This is now stated on lines 425 - 427.

Finally, no, we do not have any flexure data to back up the argument that the GPSs are only impacted by the meltwater budget in a 250m radius? The data that we present in this paper are the first to *show ice shelf flexure in response to meltwater loading and unloading on any ice shelf*. Yes, all GPS are rising, but as is explained on lines 172 - 175, that is because the general trend in our study area during the melt season is that of a negative meltwater budget, i.e. there is more water leaving the area (likely flowing into the ocean) than is ponding in the area. The GPSs that rise up more quickly than the others (i.e. particularly Ring and WT GPSs 1) are doing so due to a faster rate of water unloading in their local vicinities, due to the lakes draining (Fig 2).

163-167- confusing (and long) sentence

We have broken this long sentence into two and reworded the second of the two sentences.

180-184 – confusing sentence. If rivers are helping to evacuate water causing net meltwater budget I'm not sure more meltwater will encourage ponding and positive meltwater budgets - wouldn't it just make the rivers flow faster?

Yes, the large scale river systems may help to produce a negative net meltwater budget (i.e.

as we think it currently happening on the McMIS). However, if melt rates increase significantly, ponds may increase in number and grow in size (partly due to viscoelastic relaxation of the ice, and partly due to enhanced lake bottom ablation) such that even with a large-scale river system, the net meltwater budget may still be positive due to very large meltwater production rates. This sentence has been broken into two, and reworded to better explain this (lines 220-225).

188 – potential fracture? I thought you didn't see any fractures?

We did not observe any fractures, but that doesn't mean to say that they were not present. They may have been covered in the large quantities of debris. We have removed "that we observed" from this sentence to make it clear. The sentence now reads: "Currently on the McMIS, surface lakes are not sufficiently widespread for the localized meltwater-loading induced flexure (and potential fractures) to affect more than one lake." (lines 228 - 230)

191-196 – again the justification is given for application to models. Why not make more of the modeling that you did? That would seem the most useful application of this study. Please see our earlier extensive response to this point on page 1.

356 – was the timelapse data used to validate the PDD calculation? What were the data used for?

As explained on lines 132-133 in the main text, the time lapse camera was used to produce a movie of Rift Tip Lake filling and draining (see Supplementary Movie 1). Three frames from that movie are now also shown in a new figure (Supplementary Fig. 2). The movie (and photos) are very useful as they show that: i) Rift Tip Lake drained by overflow via a stream that developed (and therefore likely did NOT drain by hydrofracture); and ii) the water depth pressure sensor was neither located in the deepest part of the Rift Tip lake, nor in the stream that drains the lake. We were not in the field when the lake drained (we were home in the US/UK), so would not be aware of this detail had the camera not recorded it.

395 – if the threshold was lowered that much does this mean other studies using the higher threshold are wrong? This could be an important point.

No it does *not* mean that other studies using the higher threshold are wrong. Unlike ours, those studies deal with *debris free* ice/water bodies.

425 – given the debris content, how much error does this add to your calculations?

Please see the sentence from lines 489 - 492 that describes the errors.

437 – what about the assumptions mentioned in lines 421-425?

We appreciate that the presence of debris will affect water depth calculations, however, we still expect negative and positive errors to cancel one another out, as shown by Pope et al 2016 (lines 489 - 492).

441 – which ablation measurements? The ones in the lake? If measured against the GPS poles do you have multiple measurements or just between installation and retrieval?

The ablation measurements are first mentioned on lines 496 - 497, and are explained a couple of lines below (lines 496 – 498), after the AWS data are described. Two measurements at each pole were taken, one when the instruments were installed, and again when the instruments were removed (in the meantime the field team were not in Antarctica). The average ablation rate (and range) that we measured is also now stated on lines 65 – 67.

438 – can you explain the PDD more fully to that it is clear how the >12 mm we/day becomes 47.5 mm we C-1 day-1

>12 mm w.e. day⁻¹ is the *average melt rate* for the ‘dirty’ stations. 47.5 mm w.e. C⁻¹ day⁻¹ is the *PDD factor* for ‘dirty’ ice. The PDD factor is based on the number of positive degree days in the melt season (which we know from our AWS data) and the total melting that occurred during that same time period (which we know from our ablation stake measurements), which therefore gives the average amount of melting that you would expect, for this location, per degree Celsius, per day. Thus, 12 mm w.e. day⁻¹ becomes 47.5 mm w.e. C⁻¹ day⁻¹ because the average daily temperature measured over the melt season must have been ~ 0.25 °C. This is a very standard procedure used to calculate PDDs (see e.g. the melt modelling review of Hock, 2005), and therefore we do not think it needs to be explained in the Methods.

492 – wouldn’t the Maxwell time be in the range of several days? As your lakes take several weeks to drain it seems that a visco-elastic model would be more useful.

We have done extensive model studies of the viscoelastic flexure (see our various papers in the reference list) and find that the relaxation-response time (which is a more apt name for it than Maxwell time) is generally longer than just the 2 days that the referee suggests. It is more likely a week to 10 days. We state in the paper that we appreciate that “a viscous response of the ice shelf will be present” in addition to an elastic one (lines 549 -550). And we also state that the elastic model solution that we present in our paper therefore represents the maximum upper bounds for flexure and stress levels (lines 550 - 555). However, we ran a simple elastic model as we did not have all the required information to confidently run a more sophisticated viscoelastic model.

Fig 1 – the lake extent is unclear from this (or any) figure so it’s hard to understand the arguments about GPS 1 at each lake having the most uplift. This makes it very difficult to understand the GPS data in Figure 2 as it’s not clear where these are located relative to the lake.

Unfortunately the lakes on the McMIS are not nearly as clearly defined as those on the Greenland Ice Sheet, or even compared to those currently on ice shelves like the George VI. However, Supplementary Figs 1 and 2, and Movie 1, help readers to better understand the appearance and behaviour of these lakes.

Given the very complicated terrain and patterns of surface ponding on this ice shelf, our aim was to help facilitate a better understanding of the GPS and water pressure data presented in Fig. 2 through our calculations of meltwater budget calculations through the melt season (Fig 4 and Supplementary Fig 3). Those meltwater budget calculations simply assume that loads/anti-loads are located in circles of radius 250 m around each GPS (though as we now state, sensitivity tests (for circles from $r = 125 - 500$ m) showed that the precise size of these circles did not affect the overall trends in the meltwater budget calculations (lines 425-427).

Fig 2 – I don’t see any downward deflection in the GPS apart from a small change in Ring GPS1 a few days before drainage. The overall trend for the GPS appear to be upward deflection. The WT and Peanut sites don’t seem to have much of a break in slope apart from WT GPS 1. I had to read to the end of the paper before realizing some data in Figure 2a and b were estimated and not recorded – this should be clarified in the caption.

Yes, the overall trend in all ice shelf vertical elevation time series is upward movement, and we explain the reasons for this in lines 172 – 177. This dominant signal likely ‘hides’ the

signals of lake filling trends to some extent. We now note in the caption that some of the data in Figs 2a and b are estimated, and how precisely this is done.

Additional references (not referred to in our manuscript)

Borstad, C. P., Rignot, E., Mouginot, J. & Schodlok, M. P. Creep deformation and buttressing capacity of damaged ice shelves: theory and application to Larsen C ice shelf. *The Cryosphere* 7, 1931–1947 (2013).

Hock, R. (2005). Glacier melt: a review on processes and their modelling. *Progr. Phys. Geogr.*, 29(3), 362–391.

Kellogg, T., D. Kellogg and M. Stuiver (1991), Radiocarbon dates from the McMurdo Ice Shelf, Antarctica: implications for debris band formation and glacial history. *Antarct. J. US*, 26(5), 77-79.

MacAyeal, D. R., and R. H. Thomas, 1985. The effects of basal melting on the present flow of the Ross Ice Shelf. *Journal of Glaciology*, 32(110), 72-86.

Reviewers' comments:

Reviewer #1 (Remarks to the Author):

I feel the authors have replied robustly but fairly and have only one outstanding potential issue: that of GPS uncertainty. The response to reviewer letter states that this is negligible and presented in methods (pointing to lines 489 - 492 [which is not correct]). As I read it, the relevant methods section is extensive and detailed but does not dedicate argument to deriving uncertainty. If I am correct, the only statement on this is (lines 375 - 379) as follows:

"Applying the time-averaging and tide removal algorithms described above to the elevation time series presented in Fig. 2 renders errors due to short-term (seconds to minutes) influences (due to GPS receiver effects and data processing uncertainties, and to ice-shelf movements due to long-period ocean swell) to < 10 mm; and renders errors due to long-term (hours to days) effects (due to ocean tide and the IBE) to < 50 mm."

I make the following points:

First, I believe the derivation of this uncertainty should be split into its individual contributing components and analysis of each presented formally.

Second, I cannot see how there is one error (of a given fixed length) for 'seconds to minutes' and another (of a different fixed length) for 'hours to days'. This needs explaining. This implies the error increases five fold in the 1 second from 59 minutes and 59 seconds to 60 minutes and zero seconds. Surely there is no actual basis for this.

Third, how is an error '< * mm'? Does this mean '+/- * mm'? How is this error defined?

Fourth, what is the uncertainty associated with the physical mounting of the GPS antennae on their pole? If it is considered to be zero then the poles must not have swayed in the wind or shifted vertically in some other way. Fine if they were robust enough, but I believe this needs to be stated.

Fifth, ten 'abnormal' periods were removed on the basis of visual inspection. I don't really have a problem with this as a form of quality control, but were there any 'quasi outliers' or excursions that were considered real and not removed? What was the quantitative threshold used to divide the two groups? It would be nice to see the nature of the *most ambiguous* of those removed in the supplementary material.

Sixth, an uncertainty (even over hours to days) of 50 mm may well not be negligible - and I (still) believe the formally-calculated uncertainty value should be added to the relevant time series in Fig. 2. If in many instances it is considered to be smaller than the line width then good - just state so.

Reviewer #2 (Remarks to the Author):

The authors have addressed the few minor comments I had in the first round. Apologies for misunderstanding one of the methods.

Reviewer #3 (Remarks to the Author):

I have read the new version of the manuscript and the rebuttal to my original comments. Unfortunately I don't believe that the authors have significantly improved their manuscript as

many of the suggestions by myself and the other reviewers were not addressed or included in the new version of the manuscript. Although the science presented in this manuscript seems legitimate and will be of interest to glaciologists examining ice shelf ponding, it is unlikely to be of interest to a wider audience. I therefore do not think that the presented data and modeling are significant enough for publication in this journal; instead they would be appropriate for a journal more specifically directed towards glaciologists.

In the original review, both myself and another reviewer pointed out that the modeling in the paper was not extensive and the field data was not used to discuss the relevance of the model and the parameterization for wider application. This has not been addressed and I would say is the biggest limitation of this paper. The response of the authors to our suggestion that the model deserved more attention was to move the material already in the supplementary material and the methods to the main manuscript, but not to add any additional analysis. This does not address my original questions about the modeling exercise, which require much further exploration of the parameters to determine the usefulness of this exercise.

The stated aim of the authors is to show that flexure happens (ok) and apply this to larger scale ice sheet models. However, because the model was only discussed in one or two paragraphs it's not at all clear how the latter would be achieved. The model has been tuned to match the flexure observed with your GPS (although only Ring GPS 1) but how would this be applied to other areas without in situ data? Why was the model not directly compared to each set of lake data rather than a bulk comparison since the uplift rates were different at each site? Is there some relationship with the ice in that area – thickness, temperature, debris content etc. that could explain the parameterization and therefore make it more widely applicable? What can these parameters tell us about other ice shelves and, if the authors insist on continuing to focus on hydrofracture, the future hydrofracture potential of this and other ice shelves?

Without this type of analysis, the novelty and usefulness of this study is limited to a test scenario of a small area of a thin ice shelf and I don't see how it can be applied to other areas. The authors argue that applying their data in a more general sense to ice shelf models is beyond the scope of this paper and that may well be the case. However, in its current form I don't think there is enough of general interest and wider applicability in this paper to suggest publication in this journal.

There is still a strong focus on ice flexure promoting fracturing and, as I pointed out in the first version, this is misleading given that the authors did not see any fracture as a result of flexure. For example, the authors state: "Currently, the locations of fractures in these larger-scale ice-sheet models are simply parameterized according to the divergence of the ice velocity field, with fractures deepening as they fill with meltwater (i.e. through "hydrofracture")^{18,19}. Fractures in such models are not able to form as a result of ice-shelf flexure." There is no evidence that fractures do form as a result of ice-shelf flexure in this study and therefore there has been no progress on determining causes of hydrofracture or the ability to include it in larger scale process models.

The description of the GPS data in Figure 2 is opaque and at occasions misleading. In the first round of reviews, I asked for elaboration on the general upward trend of the GPS signals with the authors' response being "This dominant signal likely 'hides' the signals of lake filling trends to some extent." That is not at all clear from the description of the GPS data in line 105-142 and does this not also detract from the analysis of the flexure due to lake drainage? Also, I continue to object to the authors using 'loose sensor data' in your plots since neither of these sensors worked and all you have is a measured water depth at retrieval time for Ring, which has nothing to do with the sensor. This could be re-labelled as 'estimated lake bottom ablation' but to suggest it's estimated from transducer measurements is misleading.

Specific comments

30 – you say the shelf will flex significantly – if this is the first measurement of such an event and the justification for the study is to test this assumption, this shouldn't be written here as if it is already known.

108 - "Data from GPS 1 at all lake sites apart from the Peanut site (where we have no measured water depth data near to GPS 1), also show that there is a clear temporal coincidence between maximum water filling rates and maximum downward ice deflection rates (Fig. 2)." I see this in Ring GPS 1 at ~20/12 but not for Rift 1 or for WT GPS 1's as you state.

Fig 2 - Why is there only downward deflection at Ring GPS1 once the lake is more than 1.5m deep? If signals are present in the other GPS they are not visible in this figure in its current format.

Fig 2 - In general it's hard to see the trends from Figure 2, particularly to determine if there's any change in upward deflection once the lake has drained. A rate of change figure would be much more instructive for the readers.

111 - "There is an even clearer coincidence between times of maximum water drainage rates and upward ice movement rates at each GPS 1." Yes again for Ring GPS 1. Not for WT as the rate seems very similar to that after no lake water is present. At Rift tip, you can't say this because you've interpolated your data.

144-146: "Compared to the GPS 1 station data, which record pronounced vertical movement at the centre of all lake sites, GPSs 2 and 3 are distal from the lake centres and do not show a pronounced response to lake filling or drainage" GPS 2 and 3 at Rift Tip show an increase in elevation that is similar to the estimated one you've interpolated for GPS 1. This appears to contradict your statement that flexural responses are local.

147 - How far apart are your GPS located? You have them in Figure 1 but the scale means it's difficult to see how far beyond your modeled limit of 250m these are located, especially for Rift Tip.

200-202 – It's hard to understand the usefulness of this exercise unless the net meltwater budget is plotted directly against the GPS records. At the moment these statements just tell us there was ponding on December 24th and melting on January 18th, which in itself isn't particularly interesting. It's also not clear from Supp Fig 3 that "for both the Rift Tip and WT GPS transects, the seasonal changes in net meltwater budgets are smaller, and more consistent, across each transect, than they are at the Ring and Peanut sites." The change in WT seasonal net budget looks to be around double of Peanut and Ring and at Rift Tip is about three times as much.

434 - Why didn't you test your remote sensing lake depths against your measurement data of lake depth? It seems a little redundant to do this remote sensing analysis when you were at the field site taking depth measurements and also have time lapse imagery.

498 - In response to my previous comments the authors did not answer why the time lapse imagery was not used to validate the PDD calculations.

Response to reviews for “Direct Measurements of Ice-Shelf Flexure caused by Surface Meltwater Ponding and Drainage”

We are very grateful for the useful and constructive comments that the reviewers have provided during this second round of revisions. In particular, we are pleased to see that Reviewer 1 is now happy with every aspect of our paper (including now, the modelling component) apart from the issue of the GPS errors/uncertainties, which we have addressed below. Similarly, we are very pleased that Reviewer 2 is now entirely happy with the paper. Reviewer 3 still has concerns about the modelling aspect of the paper and has raised a few other points and made other suggestions for improvement, all of which we have addressed below.

Please note that our responses to the three reviewers' comments (below) are in blue, and line numbers refer to our revised manuscript *without track changes* (though we have also included a version of our revised manuscript *with track changes* for review).

Reviewer #1 (Remarks to the Author):

I feel the authors have replied robustly but fairly and have only one outstanding potential issue: that of GPS uncertainty. The response to reviewer letter states that this is negligible and presented in methods (pointing to lines 489 - 492 [which is not correct]). As I read it, the relevant methods section is extensive and detailed but does not dedicate argument to deriving uncertainty. If I am correct, the only statement on this is (lines 375 - 379) as follows:

"Applying the time-averaging and tide removal algorithms described above to the elevation time series presented in Fig. 2 renders errors due to short-term (seconds to minutes) influences (due to GPS receiver effects and data processing uncertainties, and to ice-shelf movements due to long-period ocean swell) to < 10 mm; and renders errors due to long-term (hours to days) effects (due to ocean tide and the IBE) to < 50 mm."

We apologise that we referred to the wrong lines in the Methods in our last response to this reviewer's comments (we accidentally referred to the details of the water depth calculations from Landsat, instead of the GPS error section). We previously described the errors as 'negligible' in the sense that they cannot (and do not) affect our overall data trends, and therefore our conclusions. However, please see below for more detail about our error calculations.

I make the following points:

First, I believe the derivation of this uncertainty should be split into its individual contributing components and analysis of each presented formally.

Second, I cannot see how there is one error (of a given fixed length) for 'seconds to minutes' and another (of a different fixed length) for 'hours to days'. This needs explaining. This implies the error increases five fold in the 1 second from 59 minutes and 59 seconds to 60 minutes and zero seconds. Surely there is no actual basis for this.

We apologize for not previously describing the degree to which our data constrain the conclusions of the manuscript in sufficient detail. The referee is quite right in pointing out the inadequacy of our 1-sentence description of uncertainty. Following the referee's suggestion that "derivation of uncertainty should be split into individual contributions and each analysed

formally”, we set out below what we believe are the key sources of uncertainty, and address them. We also indicate below the changes we have made to the manuscript and Methods.

Mechanical Problems: Antennae were attached to aluminium poles that were securely fixed into the sub-surface ice reference frame, and we did not observe the poles to tilt significantly or to sink into the ice (please see lines 391 - 393 of the Methods). While we cannot quantify our confidence that pole movement did not introduce errors, we can be qualitatively assured that errors were not introduced because no stations exhibited sudden changes of vertical elevation at the onset of melting, as determined from the AWSs operating in the area. Stations were installed when field conditions were well below freezing and so the poles froze into the holes, and all poles had tilted by when we retrieved them from the field. A posteriori, we estimate that if we had seen a pole tilt by more than 10° , we would have reported the station data corrupted. Such tilt over a 3 m pole length would represent a 4.5 cm change in vertical elevation of the antenna.

GPS Error Sources: There are myriad sources of error in the determination of position from GPS data (e.g., satellite orbit uncertainty, ionospheric effects, clock errors, instrumental phase delay, etc.), and these are exhaustively described in textbooks. These errors are lumped together in what we take to be the “raw position data” that is output from the MIT GAMIT/GLOBK/TRACK software. Some researchers cite this error to be sub-centimetre over static GPS deployments such as ours. To estimate our raw GPS vertical displacement error, we windowed our data into 1-hour windows and computed the vertical displacement difference between two stations (Ring GPS 1 and 3). The standard deviations of these vertical displacement differences within each window (over the 81-day observation period) were averaged and found to be 48 mm. The standard deviations of a single station’s (Ring GPS 1) vertical displacement within each window were averaged and found to be 21 mm. The average of 1-hour windowed standard deviations of differential vertical displacement between the two stations that differed the most (48 mm) is the basis of our understanding that GPS errors are limited to $\sim \pm 50$ mm. And therefore, assuming the 24-hour moving average implies a reduction of standard deviation in the derived quantity by $1/(\sqrt{24}) \approx 1/5$, the error estimate of the plotted data time series in Figure 2 should be about ± 10 mm. The explanation and quantification of this error estimate is now described in more detail from lines 420 – 432.

Post-Processing Analysis Errors: We process our raw GPS data by eliminating tide, correcting for the inverse barometer effect (IBE) and removing spurious cycle-slip effected data using a heuristic visual scheme. Since errors that are common to all stations have no bearing on our ability to detect flexure by comparing vertical displacements between two or more stations, error in the IBE correction has no bearing on our conclusions. This is also true for the elimination of the tide (by the least-squares process discussed in the Methods – line 407) to the extent that the tide is relatively large scale in McMurdo Sound, and there should be relatively little difference between stations located within our ~ 40 km² field area. We find that the elimination of tide from our station data suggests that there could be < 20 mm of tidal amplitude difference in the vertical displacement data among the stations. The fact that the tides are periodic with a time scale (24 hours) shorter than the multiple-week time scale of the flexure, we interpret from our data that a 20 mm error in removing tide from the stations will not be a factor affecting our conclusions. Removal of spurious cycle slips from the raw data (as described in the Methods, lines 400 - 402) was done for cosmetic reasons, to eliminate visually obvious spikes from graphs of our raw data (none of which are shown in the manuscript). If we had *not* removed the cycle slips, the results and graphs in the manuscript would be the same within limits of visual perception. The data we archive, as

cited in the manuscript, represents our best effort to eliminate known error, hence reference to the cycle-slip correction and removal is retained in the present manuscript. Because the cycle slips lasted on the order of minutes, and had amplitudes typically variable, on the order of 100's of mm, we estimate that there would be < 10 mm of influence on the long-term (24-hour running mean) vertical displacement data. The above information is summarized in the Methods (lines 400 – 419).

Unresolved Natural Processes: Peanut GPS 1 was located 80 m from a broadband seismometer (MacAyeal et al, *in review*). Sub centimetre motions with time period of ≤ 100 seconds, mostly associated with sea swell and impulsive waves propagating through the ice covered ocean, were observed by the seismometer, but were not co-witnessed by the GPS. This suggests that the GPS instrumentation has a low-frequency cut-off, probably on the order of minutes, where the instrumentation does not record real occurrence of ice-shelf vertical motion and flexure due to waves. Because of the short time scales involved in ocean wave motion relative to the multiple-week time scale of the inferred ice- shelf flexure among our stations, we do not believe there to be any effect of the processes unresolved by the GPS.

Third, how is an error ' $< * \text{ mm}$ '? Does this mean ' $\pm * \text{ mm}$ '? How is this error defined? Apologies for being unclear. We report our estimate of the single- σ standard deviation of errors where they are quantified. The \pm notation is correct and adjustments have been made to the manuscript.

Fourth, what is the uncertainty associated with the physical mounting of the GPS antennae on their pole? If it is considered to be zero then the poles must not have swayed in the wind or shifted vertically in some other way. Fine if they were robust enough, but I believe this needs to be stated.

The referee has detected that some field glaciologists can be excessively cavalier about detecting effects of survey marker disturbance. We address the concern about tilting of the poles above (under "Mechanical Problems"); however, we note that there is not a standardized field process or reporting procedure in the glaciological community for evaluating mechanical issues with survey markers and poles. Perhaps this should be changed. For now, we simply give our assurance that mechanical error is not spuriously leading us to wrong conclusions.

Fifth, ten 'abnormal' periods were removed on the basis of visual inspection. I don't really have a problem with this as a form of quality control, but were there any 'quasi outliers' or excursions that were considered real and not removed? What was the quantitative threshold used to divide the two groups? It would be nice to see the nature of the *most ambiguous* of those removed in the supplementary material.

As discussed above (under 'Post-Processing Analysis Errors'), we could have conducted and presented the entire analysis without having made any of the corrections for abnormal periods and outliers (called 'cycle slips' in our narrative above). The corrections do not influence the conclusions of our manuscript.

Sixth, an uncertainty (even over hours to days) of 50 mm may well not be negligible - and I (still) believe the formally-calculated uncertainty value should be added to the relevant time series in Fig. 2. If in many instances it is considered to be smaller than the line width then good - just state so.

Our ± 50 mm estimate of GPS error (based on calculating standard deviation of 1-hour windowed vertical displacement difference between two stations, as discussed above under

‘GPS Error Sources’) characterizes the short-term (e.g., hours) error in the estimate of the relative vertical displacement. However, assuming that the 24-hour moving average (i.e. which is applied to the time series of GPD data in Fig. 2) implies a reduction of standard deviation in the derived quantity by $1/(\sqrt{24}) \approx 1/5$, the error estimate of the plotted data time series in Figure 2 is actually only about ± 10 mm. It therefore does not make sense to add a ± 10 mm error band to *all 12 time series of GPS* data in Fig. 2, particularly as this band would not be wider than the current line widths. This is now clarified on lines 429 – 432 of the Methods.

If the reviewer is interested in learning about our GPS error analysis in even more detail, with figures and annotated graphs, we have written and attached a document called “Error Analysis” that this reviewer should be able to view. However, the detail in that document is too much for the online Supplementary Material.

Reviewer #2 (Remarks to the Author):

The authors have addressed the few minor comments I had in the first round. Apologies for misunderstanding one of the methods.

We are very pleased to hear that this reviewer is now happy with our paper.

Reviewer #3 (Remarks to the Author):

I have read the new version of the manuscript and the rebuttal to my original comments. Unfortunately I don’t believe that the authors have significantly improved their manuscript as many of the suggestions by myself and the other reviewers were not addressed or included in the new version of the manuscript. Although the science presented in this manuscript seems legitimate and will be of interest to glaciologists examining ice shelf ponding, it is unlikely to be of interest to a wider audience. I therefore do not think that the presented data and modeling are significant enough for publication in this journal; instead they would be appropriate for a journal more specifically directed towards glaciologists.

We are very sorry to hear that this reviewer does not think that we have significantly improved our manuscript, especially given that Reviewer 1 is now happy with the paper apart from the way with deal with the GPS errors, and Reviewer 2 is now completely happy. We have gone a long way to addressing Reviewer 3’s comments below, running the analytic solution for all four lakes, altering and adding figures, and modifying large tracts of text as he/she suggests. We hope that he/she is now satisfied with our changes and responses.

As our paper presents the *first field-based data to show that ice shelf flexure occurs in response to surface water ponding and movement*, we argue that our paper *will* be of interest to the *majority* of glaciologists, and not just those interested in ice-shelf ponding, as this reviewer suggests. The paper is especially important as it presents the first observational support for the mechanism that is likely to have been dominant in the collapse of ice shelves like the Larsen B in 2002. Although previous studies have modelled this process of meltwater loading-induced fracture, no previous study has observed it. As surface melting on ice shelves increases in the coming decades, many other Antarctic ice shelves may become destabilized through this flexure mechanism (which, as results in other papers suggest, is likely to lead to ice-shelf fracture). Our observational evidence for ice-shelf flexure thus should be of interest to any scientist who has interest in the stability of Antarctica’s ice shelves and the ice sheet in general. It should also be of interest more generally to media

outlets and the general public who are interested in the overall role of Antarctica in the Earth System and how this might change in the future.

Further, this reviewer suggests in his/her next comment below that if we further increase the modelling component of the paper, it will become more of interest to a wider audience. Although we respect this opinion, we disagree with it, as papers with a strong modelling focus usually appeal to a smaller, more specialist audience.

In the original review, both myself and another reviewer pointed out that the modeling in the paper was not extensive and the field data was not used to discuss the relevance of the model and the parameterization for wider application. This has not been addressed and I would say is the biggest limitation of this paper. The response of the authors to our suggestion that the model deserved more attention was to move the material already in the supplementary material and the methods to the main manuscript, but not to add any additional analysis. This does not address my original questions about the modeling exercise, which require much further exploration of the parameters to determine the usefulness of this exercise.

In this paper, we present the exact analytic-solution to idealized elastic plate flexure for three purposes, which are to show: 1) that this analytic solution can be sensibly constrained using field data in this study for *this ice shelf*, i.e. our constrained parameter values fall within sensible ranges of values; 2) that our field data from this ice shelf make sense, i.e. given the changes in the water loads, the ice thickness, and other boundary conditions, it makes sense that we only see vertical ice shelf deflection within < 500 m of lake centres; and 3) to provide a representation of estimates of stresses involved in the observed flexure.

We would like to highlight here that this *exact analytic solution is not strictly a model*, and we apologize that we had not been clear about this before in our initial and revised manuscripts. The nature of exact analytic solutions is that they only exist for special, idealized cases (in our case: only for disk-shaped loads for an elastic plate with uniform flexural rigidity parameter and simplified boundary conditions at infinite distances from the lake). We are thus limited in conducting analysis of the analytic solution, and restricted, essentially, to what we now display in the revised manuscript that accompanies this response letter. It does *not* make sense to try to constrain parameters for application to lakes on other ice shelves where we have no data (e.g. ice thickness, temperature, existing damage, salinity (e.g. the McMIS has very high salinity) and lake basin geometry and water loading/unloading) to constrain them.

Conducting sensitivity tests to explore parameter values for application to areas of *specific* ice shelves where we have data, however, *does* make sense, and this has now been done in more detail in our revised paper where we apply the solution to *all our lakes* rather than just to one as in our original submission and first revision. We would also like to stress to the reviewer that sensitivity testing and parameter exploration *for idealized ice shelves* has already been undertaken in our previous modelling-based papers (MacAyeal and Sergienko, 2013; Banwell et al 2013, MayAyeal et al 2015; Banwell and MacAyeal 2015). As other scientists working on other ice shelves collect data, that may be used to constrain the analytic solution that we apply to data from the McMIS, then collectively, the scientific community will move towards a more complete understanding of ice shelf flexure.

By now *applying the analytic solution to all of our four lakes*, this aspect of the paper has become 'more extensive' and allows us to discuss the relevance of the analytic solution and the parameterization more widely as required by the reviewer. By doing so, we have shown

that the values of two parameters (effective ice thickness (H) and Young's Modulus (E)) constrained for Ring lake, also produce a good match between the measured and analytic solution-produced vertical deflection at the other three lake sites. As these other three sites had different volumes (and hence, weights) of meltwater unloaded from them during the season (as calculated by our net seasonal meltwater budget calculations) we treated the value of lake radius (R) for each lake as an adjustable parameter. Agreement between the simulated and measured centre-lake deflections is obtained with R values of 250 m, 175 m, and 200 m for Rift Tip, WT and Peanut lakes, respectively (lines 225 – 228). We have plotted the equivalent of Fig. 3 for Rift Tip, WT and Peanut lakes in a *new* figure: Supplementary Fig. 3. These plots show the ice-shelf deflection and the associated stresses calculated by the analytic solution, for each lake, using the optimum parameter values.

What we have written above is our opinion, and those of us who are modelers, however, can appreciate the referee's opinion that more modelling would be valuable. The reviewer may also be interested to know that a 2-D, plan-view finite-element model of the field area treating the ice-shelf as a viscoelastic plate is currently being developed and is being used to investigate the problem further. However, incorporating these new modelling results in this paper would, we think, be well beyond its current scope, and would detract from the main aim our paper, which is to clearly and competently present the field data, which are the first of their kind.

Finally, we note that although Reviewer 1 had previous concerns about the modelling aspect of our study, after the last iteration, they are now completely happy with that aspect of our paper. We hope that as we have now: i) *applied the analytic solution to all four lakes*, and ii) *explained why it is not useful to discuss the parameterizations done with reference to these four lakes on the McMIS in the context of other ice shelves*, that this reviewer will now also be happy with this section of our manuscript.

The stated aim of the authors is to show that flexure happens (ok) and apply this to larger scale ice sheet models. However, because the model was only discussed the in one or two paragraphs it's not at all clear how the latter would be achieved. The model has been tuned to match the flexure observed with your GPS (although only Ring GPS 1) but how would this be applied to other areas without in situ data? Why was the model not directly compared to each set of lake data rather than a bulk comparison since the uplift rates were different at each site? Is there some relationship with the ice in that area – thickness, temperature, debris content etc. that could explain the parameterization and therefore make it more widely applicable? What can these parameters tell us about other ice shelves and, if the authors insist on continuing to focus on hydrofracture, the future hydrofracture potential of this and other ice shelves?

The referee is mistaken to believe that we advocate using the analytic-solution-based elastic plate model presented in the manuscript as the best way to apply to large-scale ice sheet models. Our actual wording in the previous version of our manuscript regarding the study's motivation was: "...to provide a basis for improving both process-scale and continent-scale models of ice-shelf behaviour in response to surface melting." It was never our intention to improve large-scale ice sheet models as part of this study. As this sentence appears to have led to confusion (and we thank the reviewer for spotting this and pointing it out), and as we agree it is not directly relevant to the aims of the current study, we have deleted it from the paragraph about the study's motivation. And we have also removed the mention of large-scale models from the abstract. Continent-scale ice sheet models are now *only* mentioned in the final sentence of the manuscript where we state: "Ultimately, the observations presented

here provide an initial performance constraint to guide the development of regional- and continent-scale ice-sheet models^{33,34} to produce more accurate predictions of future sea-level rise following ice-shelf collapse^{5,35-37}. The process of ice-shelf flexure in response to surface meltwater load changes⁹, which may lead to fracture⁷, is not yet simulated by these models” (lines 282 – 286)

Without this type of analysis, the novelty and usefulness of this study is limited to a test scenario of a small area of a thin ice shelf and I don’t see how it can be applied to other areas. The authors argue that applying their data in a more general sense to ice shelf models is beyond the scope of this paper and that may well be the case. However, in its current form I don’t think there is enough of general interest and wider applicability in this paper to suggest publication in this journal.

We appreciate the reviewer’s sentiment; but reply that characterization and direct observation of any newly found process (in our case, meltwater-induced ice-shelf flexure) that affects the Antarctic Ice Sheet with more than just a relatively small study site is hard to do. We are sure that the referee appreciates this, but take issue that the novelty and usefulness is limited because our field area is just a “small area of a thin ice shelf”. But regardless (and as we explain more thoroughly after this reviewer’s initial comment on page 4), our paper presents the *first field-based data to show that ice shelf flexure occurs in response to surface water ponding and draining*, and we argue that our paper *will* be of interest to the majority of glaciologists and to scientists interested in the Antarctic Ice Sheet in general.

In our opinion (which is based on 25+ field seasons amongst the authors), the McMIS is the best place for now to make the observations for the reasons we give on lines 41 - 43. And although the McMIS is relatively thin, and is therefore slightly atypical, the same process of meltwater loading-induced flexure on thicker ice shelves will still occur, just with a smaller deflection magnitude for a given volume of meltwater. The thickness of the ice shelf, as is well understood through viscoelastic plate theory, can be accounted for and scaled-up from our thin ice-shelf location without much controversy.

There is still a strong focus on ice flexure promoting fracturing and, as I pointed out in the first version, this is misleading given that the authors did not see any fracture as a result of flexure. For example, the authors state: “Currently, the locations of fractures in these larger-scale ice-sheet models are simply parameterized according to the divergence of the ice velocity field, with fractures deepening as they fill with meltwater (i.e. through “hydrofracture”). Fractures in such models are not able to form as a result of ice-shelf flexure.” There is no evidence that fractures do form as a result of ice-shelf flexure in this study and therefore there has been no progress on determining causes of hydrofracture or the ability to include it in larger scale process models.

We are sorry that this reviewer thinks there is still too much of a focus on the process of ice-shelf fracture in our paper. This is a fair criticism. During our last set of revisions, we did reduce this focus by concentrating more on ice-shelf flexure, while still mentioning fracture as a possible (but very important) end result of significant flexure-induced stress. We have now further decreased the focus on fracture and increased the focus on flexure (as a precursor to fracture), which we hope now satisfies this reviewer. For example, we agree that the sentence the reviewer quotes in their comment above is misleading and not entirely relevant, so we have deleted it from the revised paper. We also now make it even clearer that the lakes in our study area do not drain by hydrofracture (as now explained first in the Abstract, where we say that they “drain by overflow and channel incision” (line 17)).

However, we believe that the manuscript *should* continue to mention fracture (and hydrofracture) as the background and justification for the field observations undertaken. Numerous modelling simulations (both for ice and other elastic and viscoelastic materials) show that flexure, and more specifically, flexure-induced stress, is likely to lead to fracture if flexure stress becomes sufficiently high. Therefore, although our study did not produce direct evidence of fracture, reference to fracture should remain in the paper because it is *fracture that results from flexure-induced stresses, and that is the direct cause of ice-shelf break-up*.

The description of the GPS data in Figure 2 is opaque and at occasions misleading. In the first round of reviews, I asked for elaboration on the general upward trend of the GPS signals with the authors' response being "This dominant signal likely 'hides' the signals of lake filling trends to some extent." That is not at all clear from the description of the GPS data in line 105-142 and does this not also detract from the analysis of the flexure due to lake drainage?

We apologise that we did not satisfactorily respond to this reviewer's request to expand upon our explanation for the observed upward trend in the GPS data. We have now done this, and explained clearly why this background GPS uplift trend 'hides' the downward GPS trend when the lakes have only relatively low volumes of water in them. The observed upward trend in all the GPS data is due to a net loss of meltwater from the ice-shelf surface in the summer, due to a dominance of melting and export, over meltwater inflow (from the surrounding area) and ponding. This upward trend is therefore a consequence of the ice shelf striving to reach hydrostatic equilibrium. Deviations from this upward trend from station to station constitute the effects of flexure due to spatially and temporally varying surface loads. We have now explained this fully in the revised manuscript; please see lines 131 – 138.

Also, I continue to object to the authors using 'loose sensor data' in your plots since neither of these sensors worked and all you have is a measured water depth at retrieval time for Ring, which has nothing to do with the sensor. This could be re-labelled as 'estimated lake bottom ablation' but to suggest it's estimated from transducer measurements is misleading.

We understand why this reviewer has a problem with the expression "estimated loose sensor data". However, 'loose sensor data' is not 'estimated lake bottom ablation'. It is the estimated water *depth above the lake bottom* (i.e. because if we had managed to retrieve the loose sensor, it would have been sitting on the lake bottom, and would have moved down the pole it was tied to as the lake bottom melted). To clarify all this, we have re-labelled the loose sensor data as depth "*relative to lake bottom*", and the fixed sensor data have been labelled as "*relative to fixed sensor*". We have also edited Figure 2's caption (lines 118 - 119) and the relevant section of the Methods accordingly (lines 447 - 466).

Specific comments

30 – you say the shelf will flex significantly – if this is the first measurement of such an event and the justification for the study is to test this assumption, this shouldn't be written here as if it is already known.

We have reworded this sentence to clarify that we know *from previous model simulations and laboratory experiments* that ice shelves flex in response to water loading/unloading: "Numerical model and laboratory simulations suggest that if meltwater is advected, causing lakes to fill and drain (resulting, respectively, in local loading and unloading of the ice-shelf surface), the shelf will flex significantly, which may lead to the formation of fractures both within and outside the lake basin^{8,11}." (lines 27 – 30) The data that we present in this study

are the first *from a field-based study* that provides evidence of this process occurring on ice shelves.

108 - “Data from GPS 1 at all lake sites apart from the Peanut site (where we have no measured water depth data near to GPS 1), also show that there is a clear temporal coincidence between max water filling rates and max downward ice deflection rates (Fig. 2).” I see this in Ring GPS 1 at ~20/12 but not for Rift 1 or for WT GPS 1’s as you state.

We thank the reviewer for this useful comment. Our first comment in response to this is to say that we have reworded the sentence so that it now no longer compares *rates* in the GPS deflection and water filling time series. We decided that we do not have enough information to do this (especially as rate of change plots did not work well/look good – please see our comments on this below for further detail). Also, the water depth data near to each GPS 1 actually just show rates in depth increase, not rates in lake filling (as we don’t know the basin bathymetries), and water depths are not necessarily proportional to water volumes, and therefore loads. Therefore we now just compare the temporal coincidence between times of increasing lake depth and times of vertical downward ice deflection (lines 96 – 105).

Second, to make the coincidence that we describe above clearer for each lake basin, we have made the following edits to Fig. 2. We have added red dots to the GPS vertical deflection time series; the first red dot on each time series indicates the highest elevation the GPS reaches between the beginning of the GPS time series and when it reaches its lowest elevation during the time series, indicated by the second red dot, and the third red dot indicates the highest elevation that the GPS reaches during the entire time series. We have also added vertical red arrows (with corresponding numbers) to each GPS time series indicating: i) the total downward vertical deflection, and rate, between the first and second red dots, and ii) the total upward deflection, and rate, between the second and third red dots. We think that these additions to Fig. 2 will make the vertical elevation trends much clearer, particularly the time periods, and magnitudes, of downward vertical deflection, which we agree were not clear previously.

Fig 2 - Why is there only downward deflection at Ring GPS1 once the lake is more than 1.5m deep? If signals are present in the other GPS they are not visible in this figure in its current format.

The lack of significant downward deflection before the lake is 1.5 m deep (i.e. before about 21st Dec) is likely because the meltwater in the lake when its depth was < 1.5 m was mostly produced *in-situ*; in other words, before about 21st Dec there was limited water flow into the lake basin from the wider area. Therefore there was almost no addition of mass (i.e. meltwater) until the lake reach 1.5 m in depth. After the lake fills to depths > 1.5 m, meltwater *does* flow in from the surrounding area, resulting in a net loading near to Ring GPS 1, which is obvious in the GPS data. This point is now made in the paper (lines 130 - 141).

Fig 2 - In general it’s hard to see the trends from Figure 2, particularly to determine if there’s any change in upward deflection once the lake has drained. A rate of change figure would be much more instructive for the readers.

We agree that vertical elevation trends in Fig 2 were hard to see and we have dealt with this by adding the red dots, arrows and numbers to indicate total vertical deflections and rates, as described in response to this reviewer’s comment (above) about line 108 (of our previous manuscript). These edits make the vertical elevation trends and magnitudes in Fig. 2 much clearer.

We did experiment with plotting rate of change plots for the GPS time series, but they looked messy and would not, we think, help the reader to interpret our data. We were also concerned about what the rate of change lines (if plotted) would represent, e.g. would they represent the rate of water movement that unloads the ice shelf and therefore its elastic response, or would it also represent the rate of viscous relaxation to that unloading? Strictly, we think they would represent neither. Therefore, we decided not to add rate of change plots to our (already very busy) Fig. 2, but hope that our new edits described above (including numbers stating vertical deflection rates between various times) will satisfy the reviewer.

111 - “There is an even clearer coincidence between times of maximum water drainage rates and upward ice movement rates at each GPS 1.” Yes again for Ring GPS 1. Not for WT as the rate seems very similar to that after no lake water is present. At Rift tip, you can’t say this because you’ve interpolated your data.

We apologise that this sentence was not clear and we have now reworded it. We actually wanted to refer to the coincidence between the time periods of GPS uplift and lake drainage, rather than describing the coincidence between their rates. We now state: “There is also a clear temporal coincidence between the initiation of vertical uplift at Ring and WT GPS 1s, and the initiation of the Ring and WT lake drainages (Figs. 2a and c). The same cannot be said at Rift Tip or Peanut due to missing GPS data at Rift Tip and a lack of water depth data at Peanut (Figs 2b and d).” (lines 101 -105)

144-146: “Compared to the GPS 1 station data, which record pronounced vertical movement at the centre of all lake sites, GPSs 2 and 3 are distal from the lake centres and do not show a pronounced response to lake filling or drainage” GPS 2 and 3 at Rift Tip show an increase in elevation that is similar to the estimated one you’ve interpolated for GPS 1. This appears to contradict your statement that flexural responses are local.

We thank the reviewer for pointing this out. The observations at the Rift Tip GPSs do not contradict our statement that flexural responses are ‘local’, as we define ‘local’ as < 500 m of lake centres, and both Rift Tip GPSs 2 and 3 are actually < 250 m of Rift Tip GPS 1 (i.e. the Rift Tip lake centre). We have reworded the relevant sentences to make this point clearer: “Compared to the GPS 1 station data, which record pronounced vertical movement at the centre of all lake sites, GPSs 2 and 3 are distal from the lake centres and generally do not show a pronounced response to lake filling or drainage (Fig. 2a). The exception to this is Rift Tip GPSs 2 and 3, which are both < 250 m of Rift Tip GPS 1 and uplift vertically at a rate that is almost as rapid as that measured at GPS 1 when Rift Tip lake drains.” (lines 199 - 201)

147 - How far apart are your GPS located? You have them in Figure 1 but the scale means it’s difficult to see how far beyond your modeled limit of 250m these are located, especially for Rift Tip.

As stated on line 479, the max. distance between any GPS station is 500 m. The min. distance between any GPS stations (Rift Tip GPS 1 & 2) is 130 m. We have added an additional scale bar to the inset on Fig. 1 so that these distances can more easily be measured.

200-202 – It’s hard to understand the usefulness of this exercise unless the net meltwater budget is plotted directly against the GPS records. At the moment these statements just tell us there was ponding on December 24th and melting on January 18th, which in itself isn’t particularly interesting.

We have now plotted all the meltwater budget data with the respective GPS and lake water depth data; please see revised Fig. 2. We have also re-written the sentences that compare the GPS and meltwater budget data so that more links are made between the two datasets. We

also now clearly highlight that it is not simply the max. and min. meltwater budgets that are important, it is the *net change in the meltwater budget* (i.e. the difference in the meltwater budgets between this max and min) through the melt season that is important (lines 185-191).

It's also not clear from Supp Fig 3 that "for both the Rift Tip and WT GPS transects, the seasonal changes in net meltwater budgets are smaller, and more consistent, across each transect, than they are at the Ring and Peanut sites." The change in WT seasonal net budget looks to be around double of Peanut and Ring and at Rift Tip is about three times as much. We agree the sentence was ambiguous and thank the reviewer for spotting it. We want to draw attention to the differences in the net water budget *between* the 3 GPS positions *within* each of the four sites here, not the differences between the four sites. We have rewritten the sentence to it clearer: "However, for both the Rift Tip and WT GPS transects, the seasonal changes in net meltwater budgets are more similar across each transect (i.e. between the three GPS stations), than they are at the Ring and Peanut sites."(lines 193-195)

434 - Why didn't you test your remote sensing lake depths against your measurement data of lake depth? It seems a little redundant to do this remote sensing analysis when you were at the field site taking depth measurements and also have time lapse imagery.

We do not think it makes sense to test our in-situ measured lake depths from the pressure transducers against the lake depths derived from Landsat imagery because the in-situ depths are *point* measurements, and the Landsat derived depths are *averaged over 900 m²* Landsat pixel areas. This large difference in the spatial resolution, combined with the extremely rough surface topography of the McMIS, means that the reviewer's idea will not be particularly instructive. As described on lines 163 - 167, the purpose of our remote sensing analysis of water depths (and therefore volumes) was to calculate changes in the meltwater budget, and therefore the loading/unloading of meltwater, in order to help explain the vertical elevation changes of each GPS station.

498 - In response to my previous comments the authors did not answer why the time lapse imagery was not used to validate the PDD calculations.

Pasted below is this reviewer's comment from the previous round of revisions (in black italics), followed by our comment (in blue italics). We note that this reviewer did not previously actually ask *why* the time data were not used to validate the PDD calculation, they just asked *whether it was used* for this purpose. But we also acknowledge that we did not *explicitly* answer this initial question although we thought we'd answered it *implicitly* with our response. To answer the question explicitly: No, we did not use the time lapse imagery to validate the PDD calculations. To answer the "why" question: It was not possible to use the time lapse data to validate the PDD calculations because we did not install any vertical length scales (i.e. stadia rods) in view of the camera during the melt season. If we had done this, such measurements would have been very point specific, given the large variations in albedo and melt rates across the surface of the McMIS, however they would have acted as additional measurement points and given us continuous data to supplement our 12 ablation stakes. We will keep this useful idea in mind when planning a future field season. The other way to obtain continuous ablation measurements of course is to use sonic rangefinders.

356 - *was the timelapse data used to validate the PDD calculation? What were the data used for?*

As explained on lines 132-133 in the main text, the time lapse camera was used to produce a movie of Rift Tip Lake filling and draining (see Supplementary Movie 1). Three frames from that movie are now also shown in a new figure (Supplementary Fig. 2). The movie (and

photos) are useful as they show that: i) Rift Tip Lake drained by overflow via a stream that developed (and therefore likely did NOT drain by hydrofracture); and ii) the water depth pressure sensor was neither located in the deepest part of the Rift Tip lake, nor in the stream that drains the lake. We were not in the field when the lake drained (we were home in the US/UK), so would not be aware of this detail had the camera not recorded it.

References not in main paper:

MacAyeal, D. R., **Banwell**, A. F. Okal, E. A. Lin, J. Willis, I. C. Goodsell, B. and Macdonald, G. J. *in review*, Diurnal Seismicity Cycle Linked to Subsurface Melting on an Ice Shelf, *Ann. Glaciol.*

Reviewers' comments:

Reviewer #1 (Remarks to the Author):

I am happy now that the authors have addressed and present the potential uncertainty in the vertical GPS data.

That said, in my initial review I asked - based on the manuscript's own rationale - for the data to be used to drive a flexure model. I now note - if I read the response correctly - that a more advanced, spatial model is being developed for a separate submission. I believe that combining the data and the model (even into two parts if necessary) would result in a stronger publication.

Response to reviews for NCOMMS-18-04953B (“*Direct Measurements of Ice-Shelf Flexure caused by Surface Meltwater Ponding and Drainage*”)

Apart from Reviewer #1’s “belief” that the paper may become “stronger” with additional modelling, which, as we explain below, is based on a false premise, we have not been requested to make any additional changes to the manuscript. Unless we are presented with evidence to the contrary, we must assume that the three reviewers are now entirely happy with our paper, given that we have made all their suggested modifications. Therefore the manuscript that we have uploaded to accompany this letter is exactly the version that we uploaded during the previous round of revisions.

Please note that our responses to the three reviewers’ comments (below) are in blue text. Reviewers’ comments are in black text. Please refer to our current manuscript for full references.

Reviewer #1 (Remarks to the Author):

I am happy now that the authors have addressed and present the potential uncertainty in the vertical GPS data.

We are pleased that this reviewer is now happy with the substantial detail that we added to the main text and to the Methods sections of our manuscript during the last round of revisions. This therefore addresses their last remaining concern they had about our paper.

That said, in my initial review I asked - based on the manuscript's own rationale - for the data to be used to drive a flexure model. I now note - if I read the response correctly - that a more advanced, spatial model is being developed for a separate submission. I believe that combining the data and the model (even into two parts if necessary) would result in a stronger publication.

In response to this comment, we first note that Reviewer #1 stated very clearly that he/she was completely happy with the modelling aspects of our paper after our first round of revisions. The reviewer is therefore undermining their own review of our manuscript after the first round of revisions. In this latest response, they have speculated about what plans we may or may not have about manuscripts that we may or may not produce in the future.

With respect, the reviewer did not read our previous response correctly. Their comment shows a fundamental misunderstanding regarding the potential for additional modelling in our current study. We hope that the following three points clarify the misunderstanding.

A) We have already developed and published an advanced viscoelastic ice-shelf flexure/fracture model (MacAyeal et al, 2015) and applied it to a theoretical situation (Banwell and MacAyeal, 2015). (NB. We have uploaded these two papers as ‘Related Manuscript Files’). Although we make this point and reference these papers in our current manuscript and last rebuttal letter, we suspect that Reviewer #1 may still not have picked up on this. A subsequent modelling study, briefly alluded to in our last rebuttal letter, would intend to constrain the viscoelastic model, but would require new field data that currently do not exist! – see point B, below).

B) It is not possible to constrain our advanced viscoelastic model with just the field data we currently have and which are presented in our current paper. Significant additional field data are required to do that. Given the wealth of experience that the two modellers on the paper have (Alison Banwell, 10 years’ experience; Doug MacAyeal 45+ years’ experience), we feel exceedingly confident in making this statement. If Reviewer #1 disagrees, we invite them to explain how the unique dataset presented in our paper could be used sensibly and realistically to constrain our advanced viscoelastic model, without the use of additional data. It is our considered

opinion, that we do not have currently nearly enough, nor the correct type of field data to constrain the many parameter values and variables in the full viscoelastic model (we currently have two grant proposals under review to NSF and NERC to collect the necessary data). To properly constrain our advanced viscoelastic model, we would require detailed measurements across a study region of spatial and temporal variations in: i) ice-shelf vertical movement using a much larger array of GPS stations around lakes than we have at present; ii) ice thickness measurements; iii) vertical ice temperature profiles; and iv) vertical snow/firn/ice density profiles.

C) The field data presented in our paper are exceedingly valuable for constraining the exact analytic solution as presented in this study, but not the advanced numerical model, and that was specifically what our NSF-funded field project (PI: Doug MacAyeal) proposed to do from the outset. That is precisely why we measured the data we did, and why we installed sets of 3 GPS stations in lines directed outwards from the centre of the four lake sites. We would also like to highlight that at the request of Reviewer #3 during the last round of revisions, we have now successfully constrained the analytic solution to all four lake sites, rather than to just one.

Finally, we would like highlight that Reviewer #1 is not saying that our paper is not strong now, just that they “believe” it could become “stronger” But, as we explain above, this is based on a false premise. Does this reviewer agree with us that our conclusions based on our unique field observations (supported by our net meltwater budget calculations and a constrained analytic solution at *all four lake sites*) are novel, strong and significant? **They are the first such field data that prove that meltwater ponding and drainage on ice shelves causes ice flexure; thought to be an extremely important process for ice-shelf stability and sea level rise.** Other studies have suggested that this process *may* occur (Banwell et al, 2014; Langley et al. 2016; Bell et al. 2017; Banwell, 2017; Kingslake et al. 2017), and have modelled it theoretically using a viscoelastic model (MacAyeal and Sergienko, 2013; Banwell et al. 2013; MacAyeal et al 2015; Banwell and MacAyeal 2015), but until now, no one has provided field data to prove it. If the reviewer can forgive the analogy, we have done the ice shelf flexure equivalent of providing empirical evidence to support the Big Bang Theory by measuring the cosmic background radiation! The theory was already out there; we have provided the evidence.

Our paper, when published, will be the *only* paper that can be cited that connects ice-shelf flexure to surface water movement/drainage using field observations. It is unique, and any future modelling and/or observational study of ice shelf instability involving surface water and hydrofracture will be hard pressed *not* to cite this paper. We expect it to be cited a lot.

Reviewer #2 (Remarks to the Author):

This reviewer was not sent our paper during this round of revisions, as they were completely happy with it after the previous round of revisions.

Reviewer #3 (Remarks to the Author):

This reviewer *only* provided confidential remarks to the editor during the last round of revisions and so we have no means of responding to them. We responded fully to their first round of comments and, as far as we are concerned, dealt with all their remaining concerns regarding our paper, most notably their request that we “perform more modelling”, which we did by constraining the analytical solution for flexure of a thick elastic plate to all four of our lake GPS data sets rather than just one, as in our original submission.